# Privacy Attacks on Image AutoRegressive Models

**Antoni Kowalczuk** [* 1]  **Jan Dubiński** [* 2 3]  **Franziska Boenisch** [1]  **Adam Dziedzic** [1]

## Abstract

Image AutoRegressive generation has emerged as a new powerful paradigm with image autoregressive models (IARs) matching state-of-the-art diffusion models (DMs) in image quality (FID: 1.48 vs. 1.58) while allowing for a higher generation speed. However, the privacy risks associated with IARs remain unexplored, raising concerns regarding their responsible deployment. To address this gap, we conduct a comprehensive privacy analysis of IARs, comparing their privacy risks to the ones of DMs as reference points. Concretely, we develop a novel membership inference attack (MIA) that achieves a remarkably high success rate in detecting training images (with a True Positive Rate at False Positive Rate = 1% of 86.38% vs. 6.38% for DMs with comparable attacks). We leverage our novel MIA to provide dataset inference (DI) for IARs, and show that it requires as few as 6 samples to detect dataset membership (compared to 200 for DI in DMs), confirming a higher information leakage in IARs. Finally, we are able to extract hundreds of training data points from an IAR (e.g., 698 from VAR-*d*30). Our results suggest a fundamental privacy-utility trade-off: while IARs excel in image generation quality and speed, they are *empirically* significantly more vulnerable to privacy attacks compared to DMs that achieve similar performance. We release the code at https://github.com/sprintml/privacy_attacks_against_iars for reproducibility.

---

[*]Equal contribution [1]CISPA Helmholtz Center for Information Security, Germany [2]Warsaw University of Technology, Poland [3]NASK National Research Institute, Poland. Correspondence to: Antoni Kowalczuk , Jan Dubiński <jan.dubinski.dokt@pw.edu.pl>, Franziska Boenisch <boenisch@cispa.de>, Adam Dziedzic <adam.dziedzic@cispa.de>.

*Proceedings of the 42$^{nd}$ International Conference on Machine Learning*, Vancouver, Canada. PMLR 267, 2025. Copyright 2025 by the author(s).

## 1. Introduction

The field of visual generative modeling has seen rapid advances in recent years, primarily due to the rise of Diffusion Models (DMs) (Sohl-Dickstein et al., 2015) that achieve impressive performance in generating highly detailed and realistic images. For this ability, they currently act as the backbones of commercial image generators (Rombach et al., 2022; Team, 2022; Saharia et al., 2022). Yet, recently, their performance was closely matched or further surpassed through novel image autoregressive models (IARs). Over the last months, IARs have been achieving new state-of-the-art performance for class-conditional (Tian et al., 2024; Yu et al., 2024; Li et al., 2024) and text-conditional (Han et al., 2024; Tang et al., 2024; Fan et al., 2024) generation. The crucial improvement of their training cost and generation quality results from the *scaling laws* that previously were observed for large language models (LLMs) (Kaplan et al., 2020) with which they share both a training paradigm and architectural foundation. As a result, with more compute budget, and larger datasets, IARs can achieve better performance than their DM-based counterparts.

At the same time, the privacy risks of IARs remain largely unexplored, posing challenges for their responsible deployment. While privacy risks, such as the leakage of training data points at inference time, have been demonstrated for DMs and LLMs (Carlini et al., 2021; 2023; Duan et al., 2023a;b; Hanke et al., 2024; Huang et al., 2024; Wen et al., 2024; Hayes et al., 2025), no such evaluations currently exist for IARs. As a result, the extent to which IARs may similarly expose sensitive information remains an open question, underscoring the necessity for rigorous privacy investigations in this context.

To address this gap and investigate the privacy risks associated with IARs, we conduct a comprehensive analysis using multiple perspectives on privacy leakage. First, we develop a new membership inference attack (MIA) (Shokri et al., 2017), which aims to determine whether a specific data point was included in an IAR's training set—a widely used approach for assessing privacy risks. We find that existing MIAs developed for DMs (Carlini et al., 2023; Duan et al., 2023c; Kong et al., 2023; Zhai et al., 2024) or LLMs (Mattern et al., 2023; Shi et al., 2024), are ineffective for IARs, as they rely on signals specific to their target model. We com-

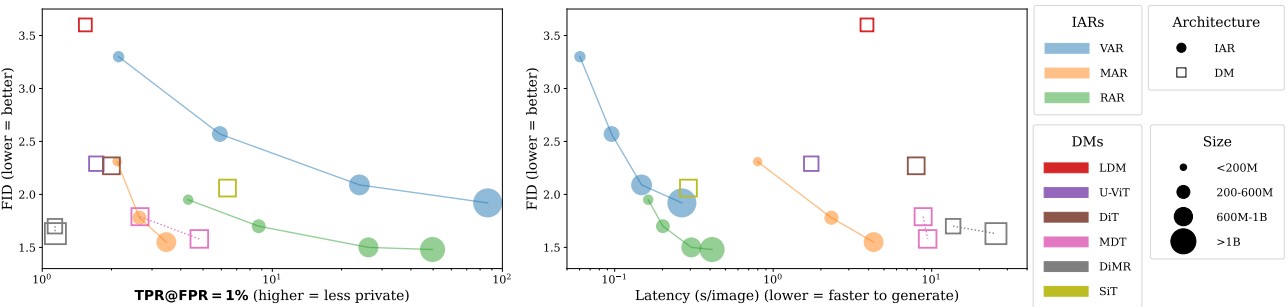

Figure 1: **Privacy-utility and generation speed-performance trade-off for IARs compared to DMs.** 1) IARs achieve better and faster image generation, but reveal more information to potential training data identification attacks. 2) In particular, large IAR models are most vulnerable. 3) In case of large IARs, even the identification of individual training samples (MIAs) has a high success rate. 4) MAR models are more private than other IARs. We attribute it to the inclusion of a diffusion module in this architecture.

bine elements of MIAs from DMs and LLMs into our new MIA based on the shared properties between the models. For example, we leverage the fact that IARs, similarly to LLMs, perform per-token prediction to obtain signal from every predicted token. However, while LLMs' training is fully self-supervised (e.g., by predicting the next word), the training of IARs can be conditional (based on a class or prompt) as in DMs. We exploit this property, previously leveraged for DMs (Zhai et al., 2024), and compute the difference in outputs between conditional and unconditional inputs as an input to MIAs. This approach allows us to achieve a remarkably strong performance of **86.38%** TPR@FPR=1%.

We employ our novel MIA to provide an efficient dataset inference (DI) (Maini et al., 2021) method for IARs. DI generalizes MIAs by assessing membership signals over entire datasets, providing a more robust measure of privacy leakage. Additionally, we optimize DI for IARs by eliminating the stage of MIA selection for a given dataset, which was necessary for prior DIs on LLMs (Maini et al., 2024; Zhao et al., 2025) and DMs (Dubiński et al., 2025). Since our MIAs for IARs consistently produce higher scores for members than for non-members, all MIAs can be utilized without any selection. This optimizations reduced the number of samples required for DI in IARs to as few as 6 samples, which is significantly fewer than at least 200 samples required for DI in DMs. Finally, we examine the privacy leakage from IARs through the lens of memorization (Feldman, 2020; Wen et al., 2024; Huang et al., 2024; Wang et al., 2024a;b; Hintersdorf et al., 2024; Wang et al., 2025). Specifically, we assess the IARs' ability to reproduce verbatim outputs from their training data during inference. We experimentally demonstrate that the evaluated IARs have a substantial tendency to verbatim memorization by extracting 698 training samples from VAR-$d$30, 36 from RAR-XXL, and 5 from MAR-H. These results highlight the varying degrees of memorization across models and reinforce the importance of mitigating privacy risks in IARs. Together,

these approaches form a comprehensive framework for empirically evaluating the privacy risks of IARs.

Our empirical analysis of state-of-the-art IARs and DMs across various scales suggests that IARs that match their DM-counterparts in image generative capabilities are notably more susceptible to privacy leakage. We also explore the trade-offs between privacy risks and other model properties. Specifically, we find that, while IARs are more cost-efficient, faster, and more accurate in generation than DMs, they empirically exhibit significantly greater privacy leakage (see Figure 1) measured against SOTA privacy attacks tailored against the respective model types. These findings highlight a critical trade-off between performance, efficiency, and privacy in IARs.

In summary, we make the following contributions:

- Our new MIA for IARs achieves extremely strong performance of even **86.38%** TPR@FPR=1%, improving over naive application of MIAs by up to **69%**

- We provide a potent DI method for IARs, which requires as few as **6** samples to assess dataset membership signal.

- We propose an efficient method of training data extraction from IARs, and successfully extract up to **698** images.

- IARs can outperform DMs in generation efficiency and quality but suffer **order-of-magnitude** higher privacy leakage in MIAs, DI, and data extraction compared to DMs that demonstrate similar FID.

## 2. Background and Related Work

**Notation.** We first introduce the notation used throughout the remainder of this paper:

| Symbol | Description |
|--------|-------------|
| $C, H, W, N$ | Channels, height, width, sequence length |
| $x \in \mathbb{R}^{C \times H \times W}$ | Original image |
| $\hat{x} \in \mathbb{R}^{C \times H \times W}$ | Generated image |
| $t \in \mathbb{N}^{N}$ | Tokenized image |
| $\hat{t} \in \mathbb{N}^{N}$ | Generated token sequence |

**Image AutoRegressive modeling.** Originally, Chen et al. (2020) defined image autoregressive modeling as:

$$p(x) = \prod_{n=1}^{N} p(t_n \mid t_1, t_2, \ldots, t_{n-1}), \qquad (1)$$

where $N$ is the number of pixels in the image, $t_i$ is the value of $i^{th}$ pixel of image $x \sim \mathcal{D}_{\text{train}}$ (training data), where pixels follow raster-scan order, row-by-row, left-to-right. During training, the goal is to minimize negative log-likelihood:

$$L_{AR} = \mathbb{E}_{x \sim \mathcal{D}_{\text{train}}} \left[ -\log\left( p\left( x \right) \right) \right]. \qquad (2)$$

However, learning pixel-level dependencies directly is computationally expensive. To address the issue, VQ-GAN (Esser et al., 2020) transforms the task from next-pixel to next-token prediction. First, the VQ-GAN's encoder maps an image into (lower resolution) latent feature vector, which is then quantized into a sequence of tokens, by a learnable codebook. In effect, the sequence length is short, which enables higher-resolution and high-quality generation. Then, tokens are generated and projected back to the image space by VQ-GAN's decoder. All the subsequent IARs we introduce, utilize tokens from VQ-GAN. This token-based formulation aligns image generation more closely with natural language processing. Additionally, similarly to autoregressive language models such as GPT-2 (Radford et al.), which generate text by sequentially predicting tokens, modern IARs also employ transformer-based (Vaswani et al., 2017) architectures to model dependencies between image tokens. We focus on the recent state-of-the-art IARs.

**VAR** (Tian et al., 2024) is a novel approach to image generation, which shifts the focus of traditional autoregressive learning from next-token to next-scale prediction. Unlike classical IARs, which generate 1D token sequences from images by raster-scan orders, VAR introduces a coarse-to-fine multi-scale approach, encoding images into hierarchical 2D token maps and predicting tokens progressively from lower to higher resolutions. This preserves spacial locality and significantly improves scalability and inference speed.

**RAR** (Yu et al., 2024) introduces bidirectional context modeling into IAR. Building on findings from language modeling, specifically BERT (Devlin et al., 2019), RAR highlights the limitations of unidirectional approach, and enhances training by randomly permuting token sequences and utilizing bidirectional attention. RAR optimizes Equation (2) over all possible permutations, enabling the model to capture bidirectional dependencies, resulting in higher quality generations.

**MAR** (Li et al., 2024) uses a small DM to model $p(x)$ from Equation (1), and samples tokens from it during inference. MAR is trained with the following loss objective:

$$L_{DM} = \mathbb{E}_{\epsilon, s} \left[ || \epsilon - \epsilon_{\theta} \left( t_n^s \mid s, z \right) ||^2 \right], \qquad (3)$$

where $\epsilon \sim \mathcal{N}(\mathbf{0}, \mathbf{I})$, $\epsilon_{\theta}$ is the DM, $t_n^s = \sqrt{\bar{\alpha}_s} t_n + \sqrt{1 - \bar{\alpha}_t} \epsilon$ and $\bar{\alpha}_s$ is DDIM's (Song et al., 2020) noise schedule, $s$ is the timestep for diffusion process, and $z$ is conditioning input, obtained from the autoregressive backbone, from the previous tokens. This loss design allows MAR to operate with continuous-valued tokens, contrary to VAR and RAR, which use discrete tokens. MAR also integrates masked prediction strategies from MAE (He et al., 2022), into the IAR paradigm. Specifically, MAR predicts masked tokens, based on unmasked ones, formulated as $p(x \cdot \neg M \mid x \cdot M)$, where $M \in [0, 1]^N$ is random binary mask. Like to RAR, MAR utilizes bidirectional attention during training. Its autoregressive backbone differs from other IARs, as MAR employs a ViT (Dosovitskiy et al., 2021) backbone.

**Sampling** for IARs is based on $p(x)$, which models the distribution of the next token conditioned on the previous ones in the sequence. For VAR and RAR, operating on discrete tokens, the next token can be predicted via greedy or top-$k$ sampling. In contrast, MAR samples tokens by the DM module, which performs 100 DDIM (Song et al., 2020) denoising steps. During a single sampling step, VAR outputs a 2D token map, RAR predicts a single token, and MAR generates a batch of tokens.

## 3. Privacy Evaluation Frameworks

We assess IARs' privacy risks from the three perspectives of membership inference, dataset inference, and memorization.

### 3.1. Membership Inference

Membership Inference Attacks (MIAs) (Shokri et al., 2017) aim to identify whether a specific data point was part of the training dataset for a given machine learning model. Many MIAs have been proposed for DMs (Duan et al., 2023c; Zhai et al., 2024; Carlini et al., 2023; Kong et al., 2023), but these methods are tailored to DM-specific properties and do not transfer easily to IARs. For instance, some directly exploit the denoising loss (Carlini et al., 2023), while others (Kong et al., 2023), leverage discrepancies in noise prediction between clean and noised samples. CLiD (Zhai et al., 2024) sources membership signal from the difference between conditional and un-conditional prediction of the DM. Since IARs are also trained with conditioning input, we leverage CLiD to design our MIAs in Section 5.1.

MIAs are also popular against LLMs (Mattern et al., 2023; Shi et al., 2024) where they often work with per-token logit outputs of the model. For example, Shi et al. (2024) in-

troduce the MIN-K% PROB metric, which computes the mean of lowest $k$%-log-likelihoods in the sequence, where $k$ is a hyper-parameter. Zlib (Carlini et al., 2021) leverages the compression ratio of predicted tokens using the *zlib library* (Gailly & Adler, 2004) to adjust the metric to the level of complexity of the input sequence. Hinge (Bertran et al., 2024) metric computes the mean distance between tokens' log-likelihood and the maximum of the remaining log-likelihoods. SURP (Zhang & Wu, 2024) computes the mean of log-likelihood of the tokens with the lowest $k$%-log-likelihoods in the sequence, where $k$ is some pre-defined threshold. MIN-K%++ (Zhang et al., 2024b) is based on MIN-K% PROB, but the per-token log-likelihoods are normalized by the mean and standard deviation of the log-likelihoods of preceding tokens. CAMIA (Chang et al., 2024) computes the mean of log-likelihoods that are smaller than the mean log-likelihood, and the mean of log-likelihoods that are smaller than the mean of the log-likelihoods of preceding tokens, as well as the slope of log-likelihoods. More detailed description of MIAs can be found in Appendix D.2. While LLM MIAs seem to be a natural choice for membership inference on IARs, it is completely unclear whether approaches from the language domain transfer to IARs. In our work we show that the success of this transferability is limited (see Section 5.1), hence, we design novel MIAs, by exploiting unique properties of IARs. Our methods achieve significant improvements over initial MIAs with up to **69%** higher TPR@FPR=1% compared to the baselines.

## 3.2. Dataset Inference

Dataset Inference (DI) (Maini et al., 2021) aims to determine whether a specific dataset was included in a model's training set. Therefore, instead of focusing on individual data points like MIAs, DI aggregates the membership signal across a larger set of training points. With this strong signal, it can uniquely identify whether a model was trained on a given (private) dataset, leveraging strong statistical evidence. Similarly to MIAs, DI can serve as a proxy for estimating privacy leakage from a given machine learning model: DI provides insight into how easily one can determine which datasets were used to train a model, for instance, by analyzing the effect size from statistical tests. A higher success rate in DI indicates greater potential privacy leakage.

**Previous DI Methods.** For supervised models, DI involves the following three steps: (1) obtaining specific features from data samples, based on the observation that training data points are further from decision boundaries than test samples, then (2) aggregating the extracted information through a binary classifier, and (3) applying statistical tests to identify the model's train set. This approach was later extended to self-supervised learning models (Dziedzic et al., 2022a;b), where training data representations differ from

test data, and then to LLMs (Maini et al., 2024; Zhao et al., 2025) and DMs (Dubiński et al., 2025) to identify the training datasets in large generative models. Since DI relies on model-specific properties, it is unclear how it can be applied to IARs. We propose how to make DI applicable and effective for IARs.

**Setup for DI.** DI relies on two data sets: (suspected) member and (confirmed) non-member sets. First, the method extracts features for each sample using MIAs. Next, it aggregates the features for each sample, and obtains the final score, which is designed so that it should be higher for members. Then, it formulates the following hypothesis test: $H_0$ : mean(scores of suspected member samples) $\leq$ mean(scores of non-members), and uses the Welch's t-test for evaluation. If we reject $H_0$ at a confidence level $\alpha = 0.01$, we claim that we confidently identified suspected members as actual members of the training set.

Since the strength of the t-test depends on the size of both sample sets, the goal is to reject $H_0$ *with as few samples as possible*. Intuitively, as the difference in a model's behavior between member and non-member samples increases, rejecting $H_0$ becomes easier. A larger difference also indicates greater information leakage, allowing us to use DI to compare models in terms of privacy risks. For instance, if model A allows rejection of $H_0$ with 100 samples, while model B requires 1000 samples, model A exhibits higher leakage than model B. Throughout this paper, we refer to the minimum number of samples required to reject the null hypothesis as $P$.

**Assumptions about Data.** For the hypothesis test to be sound, the suspected member set and non-member set must be independently and identically distributed. Otherwise, the result of the t-test will be influenced by the distribution mismatch between these two sets, yielding a false positive prediction.

## 3.3. Memorization

Memorization in generative models refers to the models' ability to reproduce training data exactly or nearly indistinguishably at inference time. While MIAs and DI assess if given samples were used to train the model, memorization enables extracting training data directly from the model (Carlini et al., 2021; 2023)—-highlights an *extreme* privacy risk.

In the vision domain, a data point $x$ is memorized, if the distance $l(x, \hat{x})$ from the original $x$ and the generated $\hat{x}$ image is smaller than a pre-defined threshold $\tau$ (Carlini et al., 2023). We use the same definition when evaluating our extraction attack in Section 5.3.

Intuitively, in LLMs, memorization can be understood as the model's ability to reconstruct a training sequence $t$ when given a prefix $c$ (Carlini et al., 2021). Specifically,

$t = \text{argmax}_{t' \in \mathbb{N}^N} p_\theta(t'|c)$, where $p_\theta$ is the probability distribution of the sequence $t'$, parameterized by the LLM's weights $\theta$, akin to Equation (1). This formulation states we can extract the training sequence $t$ by constructing a prefix $c$ that makes the model output $t$, with greedy sampling.

Similarly to LLMs, IARs complete an image given an initial portion of it (a prefix), which we leverage for designing our data extraction attack. In contrast, extraction from DMs can rely only on the conditioning input (class label or text prompt), which is both costly and highly inefficient, *e.g.,* work by Carlini et al. (2023) requires to generate **175M** images in order to find just 50 memorized images, and no memorization has been shown for other large DMs. In contrast, we extract up to **698** training samples from IARs by conditioning them on a part of the tokenized image, requiring only **5000** generations.

## 4. Experimental Setup

We evaluate state-of-the-art IARs: VAR-$d\{16, 20, 24, 30\}$ ($d$ = model depth), RAR-{B, L, XL, XXL}, MAR-{B, L, H}, trained for class-conditioned generation. The IARs' sizes cover a broad spectrum between 208M for MAR-B, and 2.1B parameters for VAR-$d30$. We use IARs shared by the authors of their respective papers in their repositories, with details in Appendix E. As these models were trained on ImageNet-1k (Deng et al., 2009) dataset, we use it to perform our privacy attacks. For MIA and DI, we take 10000 samples from the training set as members and also 10000 samples from the validation set as non-members. To perform data extraction attack, we use all images from the training data. Additionally, we leverage the known validation set to check for false positives.

## 5. Our Methods for Assessing Privacy in IARs

In the following we investigate privacy risks of IARs. We start from baseline, LLM-based approaches, and show how to tailor them to IARs to increase privacy leakage. As we find that IARs leak more than DMs we provide insights to explain why does it happen.

### 5.1. Tailoring Membership Inference for IARs

**Baselines.** We comprehensively analyze how existing MIAs designed for LLMs transfer to IARs. Our results in Table 1 (detailed in Appendix H ) indicate that off-the-shelf MIAs for LLMs perform poorly when directly applied to IARs. We report the TPR@FPR=1% metric to measure the true positive rate at a fixed low false positive rate, which is a standard metric to evaluate MIAs (Carlini et al., 2022). For smaller models, such as VAR-$d16$, MAR-B, and RAR-B, all MIAs exhibit performance close to random guessing ($\sim 1\%$). As model size and the number of parameters

increase, the membership signal strengthens, improving MIAs' performance in identifying member samples. Even in the best case (CAMIA with TPR@FPR=1% of 16.69% on the large VAR-$d30$), the results indicate that the problem of reliably identifying member samples remains far from being solved. These findings align with results reported for other types of generative models, as demonstrated by Maini et al. (2024); Zhang et al. (2024a); Duan et al. (2024) in their evaluation of MIAs on LLMs and by (Dubiński et al., 2024; Zhai et al., 2024) for DMs, where the utility of MIAs for models trained on large datasets was shown to be severely limited.

**Our MIAs for VARs and RARs.** To provide powerful MIAs for IARs, we leverage the models' key properties. Specifically, we exploit the fact that IARs utilize classifier-free guidance (Ho & Salimans, 2022) during training, *i.e.,* in the forward pass, images are processed both with and without conditioning information, such as class label. This distinguishes IARs from LLMs, which are trained without explicit supervision (no conditioning). Consequently, MIAs designed for LLMs fail to take advantage of this additional conditioning information present in IARs. We build on CLiD (Zhai et al., 2024), and compute $p(x|c) - p(x|c_{null})$, where $c$—class label, $c_{null}$—null class, and use this difference as an input to MIAs, instead of per-token logits. We differ from CLiD in the following way: (1) Our method works directly on $p(x)$, whereas CLiD uses model loss to perform the attack. (2) Our attack is parameter free—CLiD requires hyperparameter search and a set of samples to fit a Robust-Scaler to stabilize the MIA signal. We provide a more generalized approach, moreover our results in Table 1 demonstrate even up to a **69.69%** increase in the TPR@FPR=1% for the VAR-$d30$ model.

**Our MIAs for MARs.** Many MIAs for LLMs (Hinge, MIN-K%++, SURP) require logits to compute their membership scores. However, we cannot apply these MIAs to MAR since MAR predicts continuous tokens instead of logits. We instead use per-token loss values obtained from Equation (3) to adapt other LLM MIAs (Loss, Zlib, MIN-K% PROB, CAMIA). As the tokens for MAR are generated using a small diffusion module, we can apply insights from MIAs designed for DMs and target the diffusion module directly in our attack. We detail our MIA improvements for MAR, which counter randomness from the diffusion process and binary masks.

*Improvement 1: Adjusted Binary Masks.* MAR extends the IAR framework by incorporating masked prediction strategies, where masked tokens are predicted based on visible ones. We hypothesize that adjusting the masking ratio during inference can amplify membership signals. We increase this parameter from 0.86 (training average) to 0.95, which improves MIA and suggests that an optimal masking

Table 1: **Performance of our MIAs vs baselines.** We report the standard TPR@FPR=1% for best MIAs per model. *Baselines* refers to a unmodified naive application of LLM-specific MIAs to IARs.

| Model | VAR-*d*16 | VAR-*d*20 | VAR-*d*24 | VAR-*d*30 | MAR-B | MAR-L | MAR-H | RAR-B | RAR-L | RAR-XL | RAR-XXL |
|---|---|---|---|---|---|---|---|---|---|---|---|
| Baselines | 1.62 | 2.21 | 3.72 | 16.68 | 1.69 | 1.89 | 2.18 | 2.36 | 3.25 | 6.27 | 14.62 |
| Our Methods | **2.16** | **5.95** | **24.03** | **86.38** | **2.09** | **2.61** | **3.40** | **4.30** | **8.66** | **26.14** | **49.80** |
| Improvement | **+0.54** | **+3.73** | **+20.30** | **+69.69** | **+0.40** | **+0.73** | **+1.22** | **+1.94** | **+5.41** | **+19.87** | **+35.17** |

rate exposes more membership information.

*Improvement 2: Fixed Timestep.* Carlini et al. (2023) reported that MIAs on DMs perform best when executed for a specific denoising step $t$. Since tokens in MAR are generated using a small diffusion module, we can take advantage of this by executing MIAs at a fixed timestep $t$ rather than a randomly chosen one. Interestingly, we find that $t = 500$ is the most discriminative, differing from the findings for full-scale DMs, for which $t = 100$ gives the strongest signal Carlini et al. (2023).

*Improvement 3: Reduced Diffusion Noise Variance.* The MAR loss in Equation (3) exhibits high variance due to its dependence on randomly sampled noise $\epsilon$. To mitigate this, we increase the noise sampling count from the default 4 used during training to 64, computing the mean loss to obtain a more stable signal.

More detailed description of these improvements can be found in Appendix G. Our results in Table 2 highlight the importance of our changes to evaluate MAR's privacy leakage correctly. Thanks to our improved MIAs we do not under-report the privacy leakage they exhibit.

Table 2: Ablation of improvements to MAR MIAs. Each modification further strengthens the membership signal. We report TPR@FPR=1% values and gains.

| Method | MAR-B | MAR-L | MAR-H |
|---|---|---|---|
| Baseline | 1.69 | 1.89 | 2.18 |
| + Adjusted Binary Mask | 1.88 (+0.19) | 2.25 (+0.36) | 2.88 (+0.70) |
| + Fixed Timestep | 1.88 (+0.00) | 2.41 (+0.17) | 3.30 (+0.42) |
| + Reduced Noise Variance | **2.09 (+0.21)** | **2.61 (+0.20)** | **3.40 (+0.10)** |

**Overall Performance and Comparison to DMs** We present our results in Figure 1, evaluate overall privacy leakage and compare IARs to DMs based on the TPR@FPR=1% of MIAs. For DMs we use the strongest attack available at the time of writing this paper—CLiD (Zhai et al., 2024). In general, smaller and less performant models exhibit lower privacy leakage, which increases with model size. Notably, VAR-*d*30 and RAR-XXL achieve TPR@FPR=1% values of 86.38% and 49.80%, respectively, indicating a substantially higher privacy risk in IARs compared to DMs. In contrast, the highest TPR@FPR=1% observed for DMs is only 6.38% for SiT-XL/2 (see also Table 18).

**Possible Reasons Behind Higher Leakage of IARs** With

IARs emerging as a less private alternative to DMs, we investigate the causes behind that phenomenon. First, we ask if IARS inherently leak more because of their design. We identify three key characteristics of IARs that cause greater leakage: (1) Access to $p(x)$—IARs expose it at the output, contrary to DMs. (2) AutoRegressive training exposes IARs to more data per update. (3) Each token predicted by an IAR leak unique information about the model, amplifying leakage. We provide more details in Appendix A.1. Next, we scrutinize architecture-agnostic causes of leakage: training duration, and model size. Our results in Table 5 in Appendix A.2 show that indeed, these two factors correlate with the leakage metrics. Interestingly, for IARs the vulnerability differs with model size, while for DMs—with training duration. We also test a binary factor "Is IAR" (1 if the model is IAR, 0 otherwise), which also correlates with metrics, further confirming our intuitions about the inherent causes of leakage in IARs. We note taht MIAs are significantly less effective at identifying member samples in MARs. We attribute this to MAR's use of a diffusion loss function (Equation (3)) for modeling per-token probability, which replaces categorical cross-entropy loss and eliminates the need for discrete-valued tokenizers.

**Vulnerability of IARs Through a Lens of a Unified MIA** Finally, we look into the DM- and IAR-specific MIAs used in our study. We acknowledge that because DMs and IARs are two different classes of models, the MIAs that target each of the architectures also differ. Effectively, that variability might be the root cause of the observed discrepancy in MIA success. To evaluate that idea, we design a *Unified MIA*—an identical MIA for DMs and IARs—based on model- and architecture-agnostic Loss Attack (Yeom et al., 2018). We discard any IAR-specific improvements introduced in this section, and any DM-specific improvements from prior work (Carlini et al., 2023). Effectively, with Unified MIA we mitigate the potential influence of discrepancy in the MIA design on the final privacy assessment. Our results in Table 7 show that Unified MIA performs better than random guessing against IARs, while DMs show no leakage from that attack.

### 5.2. Dataset Inference

While our results in Table 1 demonstrate impressive MIA performance for large models (such as VAR-*d*30 with 2.1B parameters), privacy risk assessment for smaller models

Table 3: **DI for IARs.** We report the reduction in the number of samples required to carry out DI. Our improvements allow to successfully run DI on IARs even with fewer than 10 samples. *Baseline* refers to LLM DI (Maini et al., 2024).

| Model | VAR-$d$16 | VAR-$d$20 | VAR-$d$24 | VAR-$d$30 | MAR-B | MAR-L | MAR-H | RAR-B | RAR-L | RAR-XL | RAR-XXL |
|---|---|---|---|---|---|---|---|---|---|---|---|
| Baseline | 2000 | 300 | 60 | 20 | 5000 | 2000 | 900 | 500 | 200 | 40 | 30 |
| +Optimized Procedure | 600 | 200 | 40 | 8 | 4000 | 2000 | 800 | 300 | 80 | 30 | 10 |
| Improvement | -1400 | -100 | -20 | -12 | -1000 | 0 | -100 | -200 | -120 | -10 | -20 |
| +Our MIAs for IARs | **200** | **40** | **20** | **6** | **2000** | **600** | **300** | **80** | **30** | **20** | **8** |
| Improvement | -400 | -160 | -20 | -2 | -2000 | -1400 | -500 | -220 | -50 | -10 | -2 |

(such as VAR-$d$16 with 310M parameters) needs improvement. To address this, we draw on insights from previous work on DI (Maini et al., 2024; Dubiński et al., 2025), which has proven effective when MIAs fail to achieve satisfactory performance. The advantage of DI over MIAs lies in its ability to aggregate signals across multiple data points while utilizing a statistical framework to amplify the overall membership signal, yielding more reliable privacy leakage assessment. We find that while the framework of DI is applicable to IARs, its crucial parts must be improved to boost DI's effectiveness on IARs. In the following we detail these improvements.

**Improvement 1: Optimized DI Procedure.** Existing DI techniques for LLMs (Maini et al., 2024) and DMs (Dubiński et al., 2025) follow a four-stage process, with the third stage involving the training of a linear classifier. This classifier is used to weight, scale, and aggregate signals from individual MIAs, where each MIA score serves as a separate feature. This step is crucial for selecting the most effective MIAs for a given dataset while suppressing ineffective ones that could introduce false results. However, we observe that MIA features for IARs are well-behaved, meaning that, on average, they are consistently higher for members than for non-members. Thus, instead of training a linear classifier on MIA features, which requires additional auditing data, we adopt a more efficient approach: we first normalize each feature using MinMaxScaler to the [0,1] interval, and then we sum them to obtain the final per-sample score, used by the t-test. This eliminates the need to allocate scarce auditing data for training a linear classifier.

Our results for the optimized DI procedure are presented in Table 3. We observe a significant reduction in the number of samples required to perform DI for smaller models, with reductions of up to 70% for VAR-$d$16.

**Improvement 2: Our MIAs for IARs.** Our results in Table 3 indicate that as model size increases, the membership signal is amplified, enabling DI to achieve better performance with fewer samples. However, the main problem is the mixed reliability of DI when utilizing baseline MIAs as feature extractors. This issue is especially evident for smaller models, such as VAR-$d$16 and MAR-B, where DI requires thousands of samples to successfully reject the null hypothesis when the suspect set is part of the training

data. Building on the performance gains of our tailored MIAs (Table 1) we apply them to the DI framework as the more powerful feature extractors to further strengthen DI for IARs. Our improvements through stronger MIAs further enhance DI, fully exposing privacy leakage in IAR models. As a result, the number of required samples to execute DI drops to a few hundred, for example, down to only 200 for VAR-$d$16. Overall, as shown in Table 3, replacing the linear classification model with summation and transitioning to our MIAs for IARs as feature extractors significantly reduces the number of samples required to reject $H_0$.

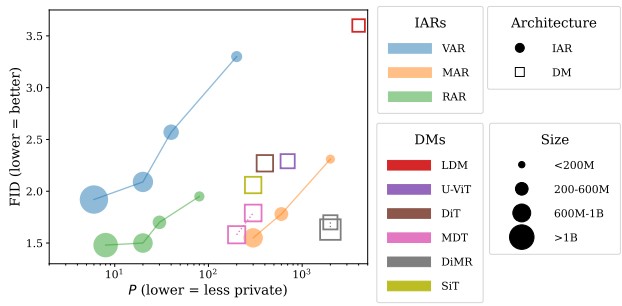

Figure 2: **DI success for IARs vs DMs.** We report the generative quality expressed with the FID score vs the number of suspect samples $P$ required to carry out DI.

**Overall Performance and Comparison to DMs.** We present our results in Figure 2, evaluating the overall privacy leakage and comparing IARs to DMs based on the number of required samples ($P$) to perform DI. Recall that a lower $P$ under the DI framework indicates greater privacy vulnerability, as it means fewer data points are needed to reject the null hypothesis—$H_0$. Our findings indicate that the same trend observed in MIAs extends to DI. Overall, models with a higher TPR@FPR=1% in Table 1 for MIAs also require smaller suspect sets $P$ for DI. Specifically, DI shows that larger models exhibit greater privacy leakage, with VAR-$d$30 and RAR-XXL being the most vulnerable. Crucially, our results clearly demonstrate that IARs are significantly more susceptible to privacy leakage than DMs. While MDT shows lower generative quality (as indicated by a higher FID score), it requires substantially more samples for DI (higher $P$ value), resulting in much lower privacy leakage.

**Why do We (Again) Observe Higher Leakage of IARs?** MIAs are the backbone of the DI framework, extracting

features from the samples to capture differences between members and non-members. When they succeed more for one class of the models, we expect that DI will also perform better for that class. With MIAs, we observe higher leakage of IARs, which stems from the increased difference between the distributions of the MIA-specific score for member and non-member samples. Because we use these scores to perform the t-test, when the difference between these distributions increase, we need a smaller $P$ to reject $H_0$. Importantly, all insights about leakage from MIAs (Section 5.1) also hold for DI. Results for correlation (Table 5) and DI performance with Unified MIA as the feature extractor (Table 7) corroborate the ones for MIA, and provide an alternative perspective into the privacy of IARs.

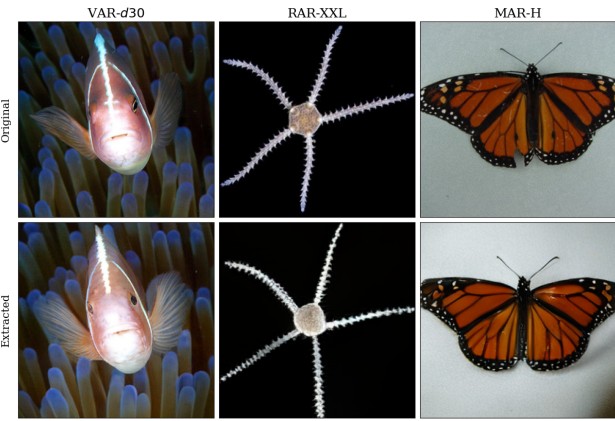

Figure 3: **Extracted Training Samples.** We note that IARs can reconstruct verbatim images from their training data. The first row shows the original training samples and the second one presents the extracted images.

### 5.3. Extracting Training Data from IARs

To analyze memorization in IARs, we design a novel training data extraction attack for IARs. This attack builds on elements of data extraction attacks for LLMs (Carlini et al., 2021) and DMs (Carlini et al., 2023). Integrating elements from both domains is required since IARs operate on tokens (similarly to LLMs), which are then decoded and returned as images (similarly to DMs). In particular, we make the observation that, on the token level, IARs exhibit a similar behavior that was previously observed for LLMs (Carlini et al., 2021). Namely, for memorized samples, they tend to complete the *correct ending* of a token sequence when prompted with the sequence's prefix. We exploit this behavior and 1) identify candidate samples that might be memorized, 2) generate them by starting from a prefix in their token space, and sampling the remaining tokens from the IAR, and finally 3) compare the generated image with the original candidate image. We report a sample as memorized when the generated image is near identical to the original image. In the following, we detail the individual building

blocks of the attack.

**1) Candidate Identification.** To reduce the computational costs, we do not simply generate a large pool of images, but identify promising candidate samples that might be memorized, before generation. Specifically, we feed an entire tokenized image $t$ into the IAR, which predicts the full token sequence $\hat{t}$ in a *single step*. Then, we compute the distance between original and predicted sequence, $d(t, \hat{t})$, which we use to filter promising candidates. This approach is efficient, since for IARs the entire token sequence can be processed at once, significantly faster than if we sampled them iteratively. For VAR and RAR we use per-token logits, and apply greedy sampling, with $d(t, \hat{t}) = 100 - \frac{100 \cdot \sum_{i=1}^{N} \mathbb{1}(t_i = \hat{t_i})}{N}$ — an average prediction error. For MAR, we sample $95\%$ of the tokens from the remaining $5\%$ unmasked in a single step, and set $d(t, \hat{t}) = ||t - \hat{t}||_2^2$, as MAR's tokens are continuous. Following the intuition that $\hat{t}$ is memorized if $\hat{t} = t$, for each model, for each class we select top-5 samples with the smallest $d$, and obtain 5000 candidates per model. Our candidate identification steps greatly improves the extraction efficiency over previous approaches (Carlini et al., 2023). We show the success of our filtering in Appendix K.3.

**2) Generation.** Then, following the methodology established for LLMs by (Carlini et al., 2021). for each candidate we select the first $i$ tokens as a prefix. The parameter $i$ is a hyperparameter and we present our best choices for the models in Table 21. We perform iterative greedy sampling of the remaining tokens in the sequence for VAR and RAR, and for MAR we sample from the DM batch by batch. We do not use classifier-free guidance during generation. We note that our method *does not* produce false positives, *i.e.,* we do not generate samples from the validation set.

**3) Assessment.** Finally, we decode the obtained $\hat{t}$ into images, and assess the similarity to the original $t$. Following Wen et al. (2024), we use SSCD (Pizzi et al., 2022) score to calculate the similarity, and set the threshold $\tau = 0.75$ such that every sample with a similarity $\geq \tau$ will be considered as memorized.

Table 4: **Count of Extracted Training Samples per IAR.**

| Model | VAR-$d$30 | MAR-H | RAR-XXL |
|-------|-----------|-------|---------|
| Count | 698 | 5 | 36 |

**Results.** In Figure 3 we show example memorized samples from VAR-$d$30, RAR-XXL, and MAR-H. We are not able to extract memorized images from smaller versions of these IARs. In Table 4 we see that the extent of memorization is severe, with VAR-$d$30 memorizing **698** images. We observe lower memorization for MAR-H and RAR-XXL, which is intuitive, as results from Sections 5.1, and 5.2 show that VAR-$d$30 is the most vulnerable to MIA and DI. Surprisingly, there is no memorization in token space, *i.e.,* $t \neq \hat{t}$,

we observe it only in the pixel space. We provide more examples of memorized images in Appendix K.1.

**Memorization Insights.** Many memorized samples follow a pattern: their backgrounds deviate from the "default" or typical scene, as shown in Figure 8 and Appendix K.1. We hypothesize that when a prefix contains part of this "unusual" background, the IAR is conditioned to reproduce the specific training image that originally featured it. Additionally, several extracted images appear as poorly executed center crops with skewed proportions—see, for instance, the wine bottle in Figure 7. These findings suggest memorization is driven by distinct visual cues in the prefix and can lead to the generation of replicas of its training data. Moreover, the same **5** samples were extracted from both VAR-$d$30 and RAR-XXL, *i.e.,* the same 5 training images are memorized by both models. One sample is memorized by both VAR-$d$30 and MAR-H (Fig. 8 and 9),suggesting some images are more prone to memorization across architectures.

Our results contrast with findings on DMs (Carlini et al., 2023), where extracting training data requires far more computation. The high memorization in IARs likely stems from their size, as VAR-$d$30 has 2.1B parameters—more than twice the number of parameters in DMs investigated in prior work. Importantly, our results also show a link between IAR size and memorization, with bigger IARs memorizing more. Scaling laws suggest that as IARs grow larger, their performance improves, but so does their tendency to memorize, making privacy risks more severe in high-capacity models.

## 6. Mitigation Strategies

Our privacy assessment methods rely on precise outputs from IARs to be effective. We exploit this insight to design defenses that mitigate privacy risks by perturbing model outputs, *e.g.,* with random noise. For VAR and RAR, we noise the logits, while for MAR, we add noise to continuous tokens after sampling. Our preliminary evaluation in Appendix J shows that the defenses are insufficient for VAR and RAR, as reducing the success of privacy attacks is achieved at the cost of substantially lower performance. In contrast, our proposed defense helps to protect MAR even more, with a relatively low drop in performance. However, MAR already exhibits the lowest success rate of the privacy attacks. This further emphasizes that leveraging diffusion techniques is a promising direction towards strong privacy safeguards for IARs, though further investigation is needed to confirm its effectiveness.

## 7. Discussion and Conclusions

IARs are an emerging competitor to DMs, matching or surpassing them in image quality at a higher generation speed. However, our comprehensive analysis demonstrates that IARs *empirically* exhibit significantly higher privacy risks than DMs, given the current state of privacy attacks against the respective model types. Concretely, we develop novel MIA for IARs that leverages components of the strongest MIAs from LLMs and DMs to reach an extremely high **86.38%** TPR@FPR=1%, as opposed to merely 6.38% for the strongest *DM-specific* MIAs in respective DMs. Our DI method further confirms the high privacy leakage from IARs by showing that only 6 samples are required to detect dataset membership, compared to at least 200 for reference DMs of comparable image generation utility. We also create a new data extraction attack for IARs that reconstructs even up to 698 training images from VAR-$d$30, while previous work showed only 50 images extracted from DMs. Our results indicate the fundamental privacy-utility trade-off for IARs, where their higher performance comes at the cost of more severe privacy leakage. We explore preliminary mitigation strategies inspired primarily by diffusion-based approaches, however, the initial results indicate that dedicated privacy-preserving techniques are necessary. Our findings highlight the need for stronger safeguards in the deployment of IARs, especially in sensitive applications.

## Impact Statement

Image autoregressive models (IARs) have rapidly gained popularity for their strong image generation abilities. However, the privacy risks that come associated to these advancements have remained unexplored. This work makes a first step towards identifying and quantifying these risks. Through our findings, we highlight that IARs *empirically* experience significant leakage of private data. These findings are relevant to raise awareness of the community and to steer efforts towards designing dedicated defenses. This enables a more ethical deployment of these models.

## Acknowledgments

This work was supported by the German Research Foundation (DFG) within the framework of the Weave Programme under the project titled "Protecting Creativity: On the Way to Safe Generative Models" with number 545047250. We also gratefully acknowledge support from the Initiative and Networking Fund of the Helmholtz Association in the framework of the Helmholtz AI project call under the name "PAFMIM", funding number ZT-I-PF-5-227. Responsibility for the content of this publication lies with the authors. This research was also supported by the Polish National Science Centre (NCN) within grant no. 2023/51/I/ST6/02854 and by Warsaw University of Technology within the Excellence Initiative Research University (IDUB) programme. We would like to also acknowledge our sponsors, who support our research with financial and in-kind contributions, especially the OpenAI Cybersecurity Grant.

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

# A. Why IARs (seem to) leak more privacy than DMs?

In the following we provide insights explaining the higher leakage observed in IARs. First, we focus on differences in architectures and models' internals. Then, we switch to explore architecture-agnostic factors like model size.

## A.1. Inherent differences between IARs and DMs

We note that DMs have inherently different characteristics than IARs, and we link them to the privacy risks they exhibit. We identify three key factors:

1. **Access to $p(x)$ boosts MIA** (Zarifzadeh et al., 2024). We note that IARs inherently expose the full information about $p(x)$ at the output (per-token logits, see Equation (1)). In contrast, DMs do not, as they learn to transform $\mathcal{N}(0, I)$ to the data distribution $q(x)$ by iterative denoising process. This difference is expressed with varying MIA designs for DMs and IARs—the former exploit the predicted noise, while the latter work with $p(x)$, by focusing on the logits. Our results confirm this premise—MAR is less prone to all privacy risks, and it does not output $p(x)$. It outputs continuous tokens, sampled from a diffusion module.

2. **AutoRegressive training exposes IARs to more data per update**. For each training sample passed through the IAR, the model "sees" $N$ different sequences to predict. Conversely, DMs only "sees" a single, noisy image. This influences two factors: a) training time of the model—DMs require to be trained two times longer than IARs, on average. b) privacy leakage—IARs are exposed to more information per each update step, which translates to increased vulnerability for privacy attacks like MIAs, DI, and data extraction. VAR outputs 10 sequences of tokens, and is less prone to MIA than RAR, which outputs 256 sequences, *e.g.,* VAR-*d*-20 vs. RAR-L (models of similar sizes).

3. **Multiple *independent* signals amplify leakage**. Previous works (Maini et al., 2024; Dubiński et al., 2025) aggregate signal from many MIAs to yield a stronger attack. Notably, each token predicted by IARs leaks unique information from the model, as it is generated from a (slightly) different prefix. Thus, per-token losses/logits that IAR-specific MIAs use, when aggregated, add up to a more informative signal, which in turn yields stronger MIAs. In contrast, DMs' outputs provide a general direction for the denoising process, and are strongly correlated. In effect, predictions at different timesteps do not provide enough *novel* information to the MIA to boost its strength.

We believe that these reasons are behind greater privacy leakage that we observe for IARs than for DMs.

## A.2. Architecture-agnostic differences between the models

The models evaluated in our work differ in many factors. Two of them, model size and training duration, are mostly architecture-agnostic, which means they are less related to the design choices of the specific models. As the efficacy of privacy attacks is directly related to these factors (Shokri et al., 2017), we want to assess if our results *really* show that IARs leak more than DMs. To this end, we collect five variables: TPR@FPR=1% (MIA), $P$ (DI metric), model size, training duration, and *Is IAR* for every model we evaluate in the paper (11 IARs, 8 DMs). For the first two (MIA, DI) we take them directly from Tables 1, 3 and 18. We obtain the model sizes from Tables 8 and 10. Training duration is expressed by a number of data points passed through the model at training, e.g., for RAR-B we have 400 epochs of ImageNet-1k train set, which amounts to $400 \times 1.27M \approx 0.5B$ samples seen. *Is IAR* factor is a 1 if the model is IAR, 0 otherwise. We take these variables and compute pairwise Pearson's correlation between them, using values for all the models.

In Table 5 we show correlations between factors (columns) and privacy metrics (rows). We identify the following insights:

1. **Training duration** is a factor that increases vulnerability for MIA and DI for DMs the most.

2. **Model size** influences leakage more for IARs than for DMs.

3. *Is IAR* factor plays the most significant role for the DI performance. It also correlates with MIA performance.

Our results show that while these two factors—model size and training duration—influence the performance of our attacks against the models, the results strengthen our notion that IARs tend to leak more privacy than IARs due to their inherent characteristics.

Table 5: **Correlation between different factors and privacy leakage.** Our results show that while the model-agnostic factors correlate with the performance, the fact that the model is IAR or not also correlates with the leakage.

|  | Architecture | Training Duration | Model Size | Is IAR |
|---|---|---|---|---|
| $P$ (DI) | IAR | 0.24 | -0.39 | |
| $P$ (DI) | DM | -0.58 | -0.32 | |
| $P$ (DI) | All | -0.04 | -0.28 | -0.46 |
| TPR@FPR=1% | IAR | 0.17 | 0.93 | |
| TPR@FPR=1% | DM | 0.31 | 0.11 | |
| TPR@FPR=1% | All | -0.2 | 0.87 | 0.38 |

# B. Limitations

We acknowledge our privacy analysis of the novel IARs, and comparison to DMs suffers from two limitations. We do not evaluate our attacks on the biggest available models (like Infinity (Han et al., 2024)) trained on massive (over 1B samples), messy datasets. Secondly, there are many factors crucial for MIA and DI performance, which differ in values between almost all the models. The following explains these issues in more detail.

### B.1. On the infeasibility of high-scale experiments on extremely big models

We do not assess how our attacks perform when applied to models trained on datasets of the scale higher than 1M samples. It may raise concerns about the scalability of the attacks and the insights they provide to the real-world applications. Unfortunately, IARs trained on bigger datasets than ImageNet-1k (Infinity (Han et al., 2024), HART (Tang et al., 2024)) do not disclose fully what their training data *exactly* is. Because of that, we are unable to perform a sound evaluation of the privacy attacks. We lack the ability to assess MIA's and DI's performance correctly, as these methods rely on two assumptions: (1) we know a part of the training data (members), (2) we have access to non-members that are *independent and identically distributed* (IID) with members. When we fail to satisfy (2) the methods would collapse to dataset detection (Das et al., 2024). Moreover, without satisfying (1) we cannot run MIA and DI at all.

While a *methodologically correct* evaluation of the cutting-edge models is out of our reach, we aim to provide more insight into text-to-images IARs, and see how much they leak. To this end, we run our attacks on VAR-CLIP (Zhang et al., 2024c), a VAR-$d$16 model trained on a captioned ImageNet-1k. Our results in Table 6 show that this model leaks significantly more data than its class-to-image counterpart of the same size. Moreover, the leakage is on a level similar to VAR-$d$20's—a model of double the size of VAR-CLIP. We argue that the increased leakage stems from the model overfitting more to the conditioning information, which is richer for textual data than for the class labels.

Table 6: **Leakage of VAR-CLIP compared to class-conditional VARs.** We observe increased privacy leakage over class-conditioned models, expressed by a stronger performance of our attacks.

| Model | TPR@FPR=1% | $P$ (DI) |
|---|---|---|
| VAR-CLIP | 6.30 | 60 |
| VAR-$d$16 | 2.18 | 200 |
| VAR-$d$20 | 5.92 | 40 |

### B.2. On the impossibility of a fully standardized experimental setup between the models

In the ideal scenario we are able to isolate only the factors inherent to the models' architecture, and consequently, are able to draw insights which design choices lead to what privacy risks. We would call such setup *standardized*, meaning that the models are *almost identical*, and differ only in factors we want to explore (like architecture). However, in reality we deal with too few models, each one being trained differently, which allows only for limited insights.

We note the models vary in the following ways:

1. **Training duration**, expressed by number of data points seen during training, *e.g.,* RAR-B sees $400 \times 1.27\text{M} \approx 0.5\text{B}$ samples. In DMs we evaluate the training duration varies between 0.21B to 1.79B samples seen, whereas IARs are trained with between 0.26B and 0.51B samples.

2. **Training objectives**. DMs minimize Equation (3), while IARs— Equation (2). Importantly, DMs minimize the expected error *over timesteps and data*, which necessitates a twice as long training duration for DMs than IARs (on average) to achieve comparable FID.

3. **Model sizes**. IARs benefit from scaling laws (Kaplan et al., 2020), and that allows them to be scaled up to sizes greater than DMs, before their performance plateaus. DMs cannot be scaled that well—the performance gains diminish faster with the increase of size. In effect, the biggest IARs we evaluate—VAR-$d30$ and RAR-XXL— are on average 2-3 times bigger than DMs. Since the size of the model impacts its vulnerability to privacy attacks, our analyses do not fully accommodate for that factor.

4. **Two stage architectures**. All models incorporate an encoder-decoder network for training and inference, *e.g.,* VQ-VAE (Esser et al., 2020). Importantly, these encoders differ between models. VAR's next-scale prediction paradigm requires training of a specialized encoder that understands how to process residual token maps, used during encoding an image to the sequence of discrete tokens. Moreover, VAR and RAR work with *discrete* tokens, *i.e.,* the encoder-decoder network additionally contains a quantizer module, which translates the continuous latent representations of the images to a 2D integer-only maps.

Unfortunately, these factors directly prohibit a *standardized comparison* of the privacy risks between DMs and IARs. We are not able to fix the training duration for all models—the generation quality of DMs would be significantly subpar than IARs (as DMs require twice the training time of IARs), and thus the results would be unsound. We incorporate the size of the models in Figures 1, 2 and 5, however, we acknowledge that the sizes vary between the models, and this limits our ability to fully disentangle this factor from the privacy results.

However, we are able to fix one factor for all the models: utility. We know the models we source are trained to the maximum of the potential each architecture allows, as we utilize models from papers that aim for exactly that—the best performance. We compare models that are the *upper boundary* of what is possible within the inherent limitations and trade-offs each architecture has to offer. We are deeply aware that privacy vs utility is a balancing act: better models tend to be less private. **Thus, our study fixes one of these parameters—utility—to be the highest possible for a given model**, and under that condition we evaluate how much it leaks. We believe our results provide strong empirical evidence that DMs constitute a Pareto optimum when it comes to image generation—they are comparable in FID, while being significantly more private than the novel IAR models.

## C. Privacy leakage under a unified attack

We acknowledge that the field of privacy attacks against image generative models like IARs or DMs is constantly evolving. Since our work aims to provide the current empirical insights into differences in privacy leakage between these architectures, we use *the strongest available* attacks to provide an upper boundary on the privacy leakage, following literature on privacy auditing (Nasr et al., 2023; Dwork, 2006).

However, IARs and DMs are two different classes of models. In consequence, the attacks we employ are *tailored* to their inherent properties, and thus the attacks vary. This might raise concerns of the following nature: what if the field progresses and a new, very potent attack is designed for DMs? Will our current empirical results hold, *i.e.,* can we *really* claim IARs leak more privacy than DMs, or is it just the current MIAs against DMs that are less powerful than for IARs?

We believe our insights in Appendix A provide reasons why IARs *inherently* leak more than DMs. To strengthen our results, we perform an *architecture-agnostic*, unified attack against all models—Loss Attack (Yeom et al., 2018).

### C.1. Loss Attack

Loss Attack is defined as follows: (1) For each sample we perform a forward pass through the model as it would be during the training (2) We compute the model loss (specific to each model) for the samples. (3) We use the losses to perform MIA (as in Appendix D.2), and we use the losses to perform Dataset Inference (see Appendix D.3).

Table 7: **Unified attack results.** We employ Loss Attack (Yeom et al., 2018), discarding any model-specific modifications that might strengthen the signal, to ensure a fair comparison between different model classes and architectures. The results strongly support our notion that IARs leak more privacy than DMs.

| Model | Architecture | $P$ (Dataset Inference) | TPR@FPR=1% (MIA) | AUC (MIA) | Accuracy (MIA) |
|---|---|---|---|---|---|
| VAR-$d16$ | IAR | 3000 | $1.50_{\pm 0.18}$ | $52.35_{\pm 0.40}$ | $50.08_{\pm 0.03}$ |
| VAR-$d20$ | IAR | 1000 | $1.67_{\pm 0.20}$ | $54.54_{\pm 0.40}$ | $50.11_{\pm 0.03}$ |
| VAR-$d24$ | IAR | 300 | $2.19_{\pm 0.20}$ | $59.56_{\pm 0.39}$ | $50.15_{\pm 0.04}$ |
| VAR-$d30$ | IAR | 40 | $4.95_{\pm 0.40}$ | $75.46_{\pm 0.35}$ | $50.32_{\pm 0.05}$ |
| MAR-B | IAR | 6000 | $1.43_{\pm 0.17}$ | $51.31_{\pm 0.30}$ | $50.48_{\pm 0.16}$ |
| MAR-L | IAR | 3000 | $1.52_{\pm 0.16}$ | $52.35_{\pm 0.30}$ | $50.70_{\pm 0.18}$ |
| MAR-H | IAR | 2000 | $1.61_{\pm 0.17}$ | $53.66_{\pm 0.30}$ | $51.07_{\pm 0.20}$ |
| RAR-B | IAR | 800 | $1.77_{\pm 0.25}$ | $54.92_{\pm 0.41}$ | $50.25_{\pm 0.06}$ |
| RAR-L | IAR | 400 | $2.10_{\pm 0.27}$ | $58.03_{\pm 0.40}$ | $50.39_{\pm 0.07}$ |
| RAR-XL | IAR | 80 | $3.40_{\pm 0.40}$ | $65.58_{\pm 0.38}$ | $50.81_{\pm 0.10}$ |
| RAR-XXL | IAR | 40 | $5.73_{\pm 0.52}$ | $74.44_{\pm 0.34}$ | $51.64_{\pm 0.19}$ |
| LDM | DM | $> 20000$ | $1.08_{\pm 0.13}$ | $50.13_{\pm 0.05}$ | $50.13_{\pm 0.11}$ |
| U-ViT-H/2 | DM | $> 20000$ | $0.85_{\pm 0.13}$ | $50.11_{\pm 0.09}$ | $50.07_{\pm 0.18}$ |
| DiT-XL/2 | DM | $> 20000$ | $0.84_{\pm 0.14}$ | $50.09_{\pm 0.05}$ | $50.15_{\pm 0.14}$ |
| MDTv1-XL/2 | DM | $> 20000$ | $0.85_{\pm 0.13}$ | $50.05_{\pm 0.05}$ | $50.08_{\pm 0.14}$ |
| MDTv2-XL/2 | DM | $> 20000$ | $0.87_{\pm 0.12}$ | $50.14_{\pm 0.05}$ | $50.16_{\pm 0.14}$ |
| DiMR-XL/2R | DM | $> 20000$ | $0.89_{\pm 0.13}$ | $49.55_{\pm 0.06}$ | $49.70_{\pm 0.14}$ |
| DiMR-G/2R | DM | $> 20000$ | $0.85_{\pm 0.12}$ | $49.54_{\pm 0.06}$ | $49.69_{\pm 0.13}$ |
| SiT-XL/2 | DM | 6000 | $0.95_{\pm 0.16}$ | $48.22_{\pm 0.26}$ | $49.97_{\pm 0.09}$ |

Loss Attack differs from MIAs against DMs in the following way: instead of fixing the timestep to the most optimal one ($t = 100$ (Carlini et al., 2023)), and averaging the loss over 5 different input noises (Carlini et al., 2023), we sample $t \sim \mathcal{U}[0, 1000]$, and compute the per-sample loss for *a single random noise*.

For MAR, we roll back the modifications to the diffusion module, explained in Appendix G. We do not fix the timestep to the most optimal one ($t = 500$), we compute the loss over 5 (default for training), instead of 64 (optimal) input noises, and we sample the masking ratio for each sample following the distribution used during training, instead of fixing it to 0.95—the optimal value.

For VAR and RAR, this attack is identical to the one in Table 14 (first row).

Since the DI framework relies on features obtained from different MIAs, we run DI only with the single feature—Loss Attack. We unify DI to be the same for DMs and IARs by removing the scoring function $s$ for DM-specific DI—CDI (Dubiński et al., 2025). In effect, the procedure is *identical* for DMs and IARs.

**C.2. IARs are empirically more prone to the unified attack than DMs**

Our results in Table 7 are consistent with the results achieved with DM- and IAR-specific attacks (Tables 1 and 3) Empirical data shows IARs are more vulnerable to MIAs and DI. Loss Attack does not yield TPR@FPR=1% greater than random guessing (1%) for DMs, whereas all IARs perform above random guessing. Moreover, with such a weak signal, DI ceases to be successful for DMs, requiring above 20,000 samples ($P$) to reject the null hypothesis (no significant difference between members and non-members), with one exception: SiT. Conversely, IARs retain their high vulnerability to DI, with the most private IAR—MAR-B—being similarly vulnerable to the least private DM—SiT.

We believe results obtained under the unified attack strengthen our message that current IARs leak more privacy than DMs.

# D. Additional Background

In the following we provide additional background on Diffusion Models used for comparison to IARs, details on MIAs, and precise definition of the DI procedure, as well as a description of the sampling strategies used by IARs during generation.

## D.1. Diffusion Models

Table 8: **DM details.** We report the training details for the DM models used in this work.

| | LDM | U-ViT-H/2 | DiT-XL/2 | MDTv1-XL/2 | MDTv2-XL/2 | DiMR-XL/2R | DiMR-G/2R | SiT-XL/2 |
|---|---|---|---|---|---|---|---|---|
| **Model parameters** | 395M | 501M | 675M | 700M | 742M | 505M | 1056M | 675M |
| **Training steps** | 178k | 500k | 400k | 2M | 6.5M | 1M | 1M | 7M |
| **Batch size** | 1200 | 1024 | 256 | 256 | 256 | 1024 | 1024 | 256 |
| **FID** | 3.60 | 2.29 | 2.27 | 1.79 | 1.58 | 1.70 | 1.63 | 2.06 |

We provide a brief overview of DMs used in our experiments. All models are class-conditioned latent DMs trained on the ImageNet dataset at 256×256 resolution. Except for LDM, all models utilize Vision Transformers (ViT) (Dosovitskiy et al., 2021) as their diffusion backbones. LDM instead employs the UNet architecture (Ronneberger et al., 2015), being a prior work. We refer the reader to the original publications for more details about their architectures and training strategies.

*LDM (Latent Diffusion Model)* by Rombach et al. (2022) first propose running diffusion in a learned latent space rather than in pixel space, using a U-Net as the denoising backbone.

*DiT-XL/2 (Diffusion Transformer)* by Peebles & Xie (2022) replaces the conventional U-Net with a ViT backbone.

*U-ViT-H/2* by Bao et al. (2023) adopts a ViT-based architecture with skip connections inspired by U-Nets. It treats image patches, class labels, and diffusion timesteps as input tokens in a unified transformer space.

*MDTv1-XL* and *MDTv2-XL (Masked Diffusion Transformer)* by Gao et al. (2023) apply a masked latent modeling strategy during training to enhance contextual learning. The model predicts missing latent tokens, improving training efficiency and sample quality. MDTv2 introduces architectural refinements that lead to further gains in fidelity and performance.

*DiMR-XL/2R* and *DiMR-G/2R* by Liu et al. (2024) propose a multi-resolution diffusion framework that processes features across different spatial scales. This design improves detail preservation and reduces distortions, especially when using large patch sizes. The models also incorporate time-aware normalization to enhance temporal conditioning.

*SiT-XL/2 (Scalable Interpolant Transformer)* by Ma et al. (2024) extends the DiT architecture with an interpolant mechanism that decouples the noise schedule from the model. This allows for greater flexibility in diffusion dynamics without architectural changes.

Besides these models, we additionally evaluate emerging DMs: LFM (Dao et al., 2023)—a flow-matching model, and DiT-MoE (Fei et al., 2024)—a mixture-of-experts DM, based on DiT (Peebles & Xie, 2022). We do not include these models for the final comparison for three reasons: (1) the released models are significantly smaller (130M parameters each) than all other models, (2) the released models achieve subpar FID scores (4.46 for LFM, unknown FID for DiT-MoE), (3) unknown details of training (number of iterations for DiT-MoE). For completeness, we perform MIA and DI, and report the values in Table 9.

Table 9: **Results for novel DM architectures.** We see the leakage is similar to the rest of DMs.

| Model | TPR@FPR=1% | $P$ (DI) |
|---|---|---|
| LFM | 1.79 | 2000 |
| DiT-MoE | 1.70 | 2000 |

## D.2. Membership Inference Attacks

MIAs attempt to identify whether a given input $x$, drawn from distribution $\mathcal{X}$, was part of the training dataset $\mathcal{D}_{\text{train}}$ used to train a target model $f_\theta$. We explore several MIA strategies under a gray-box setting, where the adversary has access to the model's loss but no information about its internal parameters or gradients. The goal is to construct an *attack* function $A_{f_\theta} : \mathcal{X} \to \{0, 1\}$ that predicts membership.

**Threshold-Based attack.** Threshold-based attack is a key method of establishing membership status of a sample. It relies on a metric such as Loss (Yeom et al., 2018) to determine membership. An input $x$ is classified as a member if value of the metric falls below a predefined threshold:

$$A_{f_\theta}(x) = \mathbb{1}[\mathcal{M}(f_\theta, x) < \gamma], \tag{4}$$

where $\mathcal{M}$ is the metric function, and $\gamma$ is the threshold.

**MIN-K% PROB Metric.** To address the limitations of predictability in threshold-based attacks, Shi et al. (2024) introduced the MIN-K% PROB metric. This approach evaluates the least probable $K\%$ of tokens in the input $x$, conditioned on preceding tokens, where $K$ is a hyperparameter, selected from $\{10, 20, 30, 40, 50\}$. By focusing on less predictable tokens, MIN-K% PROB avoids over-reliance on highly predictable parts of the sequence. Membership is determined by thresholding the average negative log-likelihood of these low-probability tokens:

$$A_{f_\theta}(x) = \mathbb{1}[\text{MIN-K\% PROB}(x) < \gamma].$$

The final value is reported for the best $K$.

**MIN-K% PROB ++.** MIN-K% PROB ++ refines the MIN-K% PROB method by leveraging the insight that training samples tend to be local maxima in the modeled probability distribution. Instead of simply thresholding token probabilities, MIN-K% PROB ++ examines whether a token forms a mode or has relatively high probability compared to other tokens in the vocabulary.

Given an input sequence $x = (x_1, x_2, \ldots, x_T)$ and an autoregressive language model $f_\theta$, the MIN-K% PROB ++ score is computed as:

$$\mathcal{S}_{\text{Min-K\%++}}(x) = \frac{1}{|S|} \sum_{t \in S} \frac{\log p(x_t|x_{<t}) - \mu_{x_{<t}}}{\sigma_{x_{<t}}}, \tag{5}$$

where $S$ consists of the least probable $K\%$ tokens in $x$, and $\mu_{x_{<t}}$ and $\sigma_{x_{<t}}$ are the mean and standard deviation of log probabilities across the vocabulary. Membership is determined by thresholding:

$$A_{f_\theta}(x) = \mathbb{1}[\mathcal{S}_{\text{Min-K\%++}}(x) \geq \gamma]. \tag{6}$$

Similarly to MIN-K% PROB, MIN-K% PROB ++ sweeps over $K \in \{10, 20, 30, 40, 50\}$, and the final result is reported for the best hyperparameter $K$.

**zlib Ratio Attacks.** A simple baseline attack leverages the compression ratio computed using the *zlib library* (Gailly & Adler, 2004). This method compares the model's perplexity with the sequence's entropy, as determined by its zlib-compressed size. The attack is formalized as:

$$A_{f_\theta}(x) = \mathbb{1}\left[\frac{\mathcal{P}_{f_\theta}(x)}{zlib(x)} < \gamma\right].$$

The intuition is that samples from the training set tend to have lower perplexity for the model, while the zlib compression, being model-agnostic, does not exhibit such biases.

**CAMIA** introduces several context-aware signals to enhance membership inference accuracy. The *slope signal* captures how quickly the per-token loss decreases over time, as members typically exhibit a steeper decline. *Approximate entropy* quantifies the regularity of the loss sequence by measuring the frequency of repeating patterns, while *Lempel-Ziv complexity* captures the diversity of loss fluctuations by counting unique substrings in the loss trajectory—both of which tend to be higher for non-members. The loss thresholding *Count Below* approach computes the fraction of tokens with losses below a predefined threshold, exploiting the tendency of members to have more low-loss tokens. *Repeated-sequence amplification* measures how much the loss decreases when an input is repeated, as non-members often show stronger loss reductions due to in-context learning.

**Surprising Tokens Attack (SURP).** SURP detects membership by identifying *surprising tokens*, which are tokens where the model is highly confident in its prediction but assigns a low probability to the actual ground truth token. Seen data tends to be less surprising, meaning the model assigns higher probabilities to these tokens in familiar contexts.

For a given input $x = (x_1, x_2, \ldots, x_T)$, surprising tokens are those where the Shannon entropy is low and the probability of the ground truth token is below a threshold:

$$S = \{t \mid H_t < \epsilon_e, \quad p(x_t|x_{<t}) < \tau_k\}, \tag{7}$$

where $H_t$ is the entropy of the model's output at position $t$, $\tau_k$ is the probability of the bottom $k\%$-th token. $k \in \{10, 20, 30, 40, 50\}$, and $\epsilon_e \in \{2, 4, 8, 16\}$ are hyperparameters. The SURP score is the average probability assigned to

these surprising tokens:

$$\mathcal{S}_{\text{SURP}}(x) = \frac{1}{|S|} \sum_{t \in S} p(x_t | x_{<t}). \tag{8}$$

Membership is determined by thresholding:

$$A_{f_\theta}(x) = \mathbb{1}[\mathcal{S}_{\text{SURP}}(x) \geq \gamma]. \tag{9}$$

The SUPR's result for the best combination of $k$ and $\epsilon_e$ is selected as the final performance.

### D.3. Dataset Inference

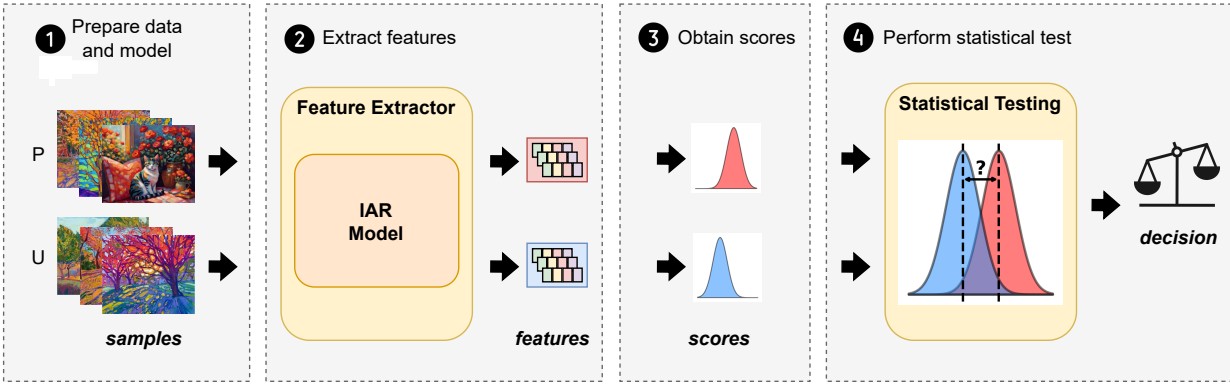

Figure 4: **Dataset Inference for IARs Procedural Steps.** The process consists of four main steps: ❶ *Data Preparation:* Prepare the data to verify whether the (suspected) member samples **P** were used to train the IAR. The (confirmed) nonmember samples **U**, from the same distribution as **P**, serve as the validation set. ❷ *Feature Extraction:* Run each individual MIA on all inputs from $\{\mathbf{P}, \mathbf{U}\}$ to extract membership features for all data samples. We use our MIAs tailored to IAR models. ❸ *Score Computation:* Normalize the extracted features using MinMaxScaler to scale them into the [0,1] range and compute a membership score for each sample by summing its normalized feature values. ❹ *Statistical Testing:* Apply a statistical t-test to verify whether the scores obtained for the public suspect data points **P** are statistically significantly higher than those for **U**. If so, **P** is marked as being part of the IAR's training set. Otherwise, the test is inconclusive and the IAR's training set is considered independent of **P**.

Scaling IARs to larger datasets raises concerns about the unauthorized use of proprietary or copyrighted data for training. With the growing adoption and increasing scale of IARs, this issue is becoming more pressing. In our work, we use DI to quantify the privacy leakage in IAR models. However, DI can be additionally used to establish a dispute-resolution framework for resolving illicit use of data collections in model training, ie. *determine if a specific dataset was used to train a IAR.*

The framework involves three key roles. First, the *victim* ($\mathcal{V}$) is the content creator who suspects that their proprietary or copyrighted data was used to train a IAR without permission. The victim provides a subset of samples ($\mathcal{P}$) they believe may have been included in the model's training dataset. Second, the *suspect* ($\mathcal{A}$) refers to the IAR provider accused of using the victim's dataset during training. The suspect model ($f_\theta$) is examined to determine whether it demonstrates evidence of having been trained on $\mathcal{P}$. Finally, the *arbiter* acts as a trusted third party, such as a regulatory body or law enforcement agency, tasked with conducting the dataset inference procedure. For instance, consider an artist whose publicly accessible but copyrighted artworks have been used without consent to train a IAR. The artist, acting as the victim ($\mathcal{V}$), provides a small subset of suspected training samples ($\mathcal{P}$). The IAR provider ($\mathcal{A}$) denies any infringement. An arbiter intervenes and obtains gray-box or white-box access to the suspect model. Using DI methodology, the arbiter determines whether the IAR demonstrates statistical evidence of training on $\mathcal{P}$.

### D.4. Sampling Strategies

The greedy approach selects the token with the highest probability. In the top-$k$ sampling, the highest $k$ token probabilities are retained, while all others are set to zero. The remaining non-zero probabilities are then re-normalized and used to determine the next token. Notably, when $k = 1$, this method reduces to greedy sampling.

# E. Model Details

In our experiments, we use a range of models from VAR (Tian et al., 2024), RAR(Yu et al., 2024), and MAR (Li et al., 2024) architectures, each varying in model size and architecture. The details of these models, including the number of parameters, training epochs, and FID scores, are summarized in Table 10. The models were trained on the class-conditioned image generation on the ImageNet dataset (Deng et al., 2009).

Table 10: **Model details.** We report the training details for IAR the models used in this work.

| | VAR Models | | | | RAR Models | | | | MAR Models | | |
|---|---|---|---|---|---|---|---|---|---|---|---|
| | VAR-$d16$ | VAR-$d20$ | VAR-$d24$ | VAR-$d30$ | RAR-B | RAR-L | RAR-XL | RAR-XXL | MAR-B | MAR-L | MAR-H |
| **Model parameters** | 310M | 600M | 1.0B | 2.1B | 261M | 462M | 955M | 1.5B | 208M | 478M | 942M |
| **Training epochs** | 200 | 250 | 300 | 350 | 400 | 400 | 400 | 400 | 400 | 400 | 400 |
| **FID** | 3.55 | 2.95 | 2.33 | 1.92 | 1.95 | 1.70 | 1.50 | 1.48 | 2.31 | 1.78 | 1.55 |

# F. Training and Inference Cost Estimation

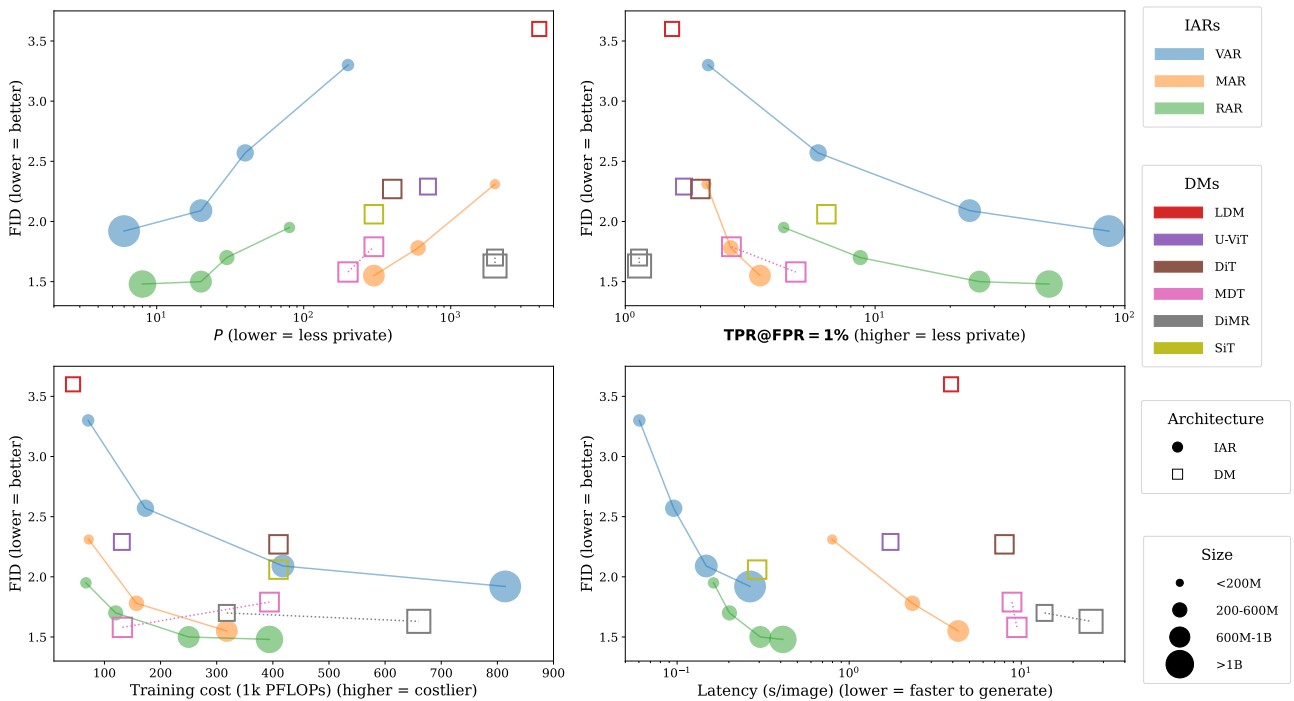

Figure 5: **Comprehensive comparison of the trade-offs between IARs and DMs.**

Here we describe the comprehensive process of training and generation cost estimation of IARs and DMs, which results in the plot Figure 5. We use *torchprofile* (tor, 2021) Python library to measure GFLOPs used for generation and training.

In order to compute the training cost, the procedure is as follows. (1) We perform a single forward pass through the model. (2) We multiply the obtained GFLOPs cost by two, to accommodate for the backward pass cost. (3) We multiply the resulting cost of a single forward and backward pass by the amount of training samples passed through the model during training. The amount of samples is based on the numbers reported in the papers for each of the evaluated models. DMs and IARs use a different reporting methodology, with the former reporting training steps and a batch size, and the latter reporting the number of epochs. For the latter, we assume that a full pass through the ImageNet-1k training set is performed, thus we multiply the number of epochs by $1,281,167$.

Time to generate a single sample (referred to as latency) is computed by generating 640 images using code from the original models' repositories. We use the maximum batch size that fits on a single NVIDIA RTX A4000 48GB GPU, to utilize our hardware to the maximum, in order to ensure a fair comparison. For DMs and IARs we follow the settings reported by

authors of the respective papers that give the lowest FID score, *i.e.,* we use classifier-free guidance for all the models. For MAR we perform 64 steps of patches sampling. For all DMs but U-ViT we perform 250 steps of denoising, while for U-ViT the reported number is 50, which explains low latency of this model in comparison to others. We acknowledge that, in case of DMs, there are ways to lower the cost of the inference, *e.g.,* by lowering the number of denoising steps. However, we use the default, yet more costly setup for these models, as there is an inherent trade-off between generation quality and cost for DMs, which we want to avoid to make our results sound.

Single generation cost in GFLOPs is computed in a similar fashion. We utilize code provided by the authors of the respective papers for the inference, wrap it using *torchprofile*, and perform a generation of a single sample. Note that here we do not measure time, and we can ignore the parallelism of hardware, as the total cost would stay the same. As we observe in Figure 1, there is a discrepancy between latency and cost of generation, especially in case of RAR, where we observe an order of magnitude higher generation time than the GFLOPs cost would suggest. This phenomenon originates from the KV-Cache mechanism that is used in case of VAR and RAR during sampling. While the compute cost is lower thanks to the mechanism, the reading operation of the cache mechanism is not effectively parallelized, which results in hardware-incurred latency. We, however, acknowledge that this trade-off might become more beneficial in cases of low-power edge devices, as the computational power of these devices is more limited than the speed of memory operations.

## G. MIAs for MAR

**Adjusting Binary Mask**   MAR extends the IAR framework by incorporating masked prediction strategies, where masked tokens are predicted based on the visible ones. This design choice is inspired by Masked Autoencoders (He et al., 2022), where selectively removing and reconstructing parts of the input allows models to learn better representations. Given that MIAs rely on detecting subtle differences in how models process known and unknown data, we hypothesize that adjusting the masking ratio during inference can amplify membership signals. By increasing the masking ratio from 0.86 (the training average) to 0.95, we create conditions where fewer tokens are available to reconstruct the original image, potentially exposing membership information more prominently.

Our experimental results, reported in Table 11, confirm that this strategy enhances MIAs' effectiveness. Specifically, TPR@FPR=1% for MAR-H increases from 2.18 to 2.88 (+0.70), and MAR-L sees an improvement from 1.89 to 2.25 (+0.36), demonstrating that a higher masking ratio strengthens membership signals. Notably, setting the mask ratio too high (e.g., 0.99) leads to a slight drop in MIA performance, suggesting a balance must be struck between revealing more membership signal and overly degrading the model's ability to generate images effectively.

Table 11: **Impact of varying mask ratio on MIAs for MAR.** We report TPR@FPR=1%. Higher values indicate stronger membership signals. The best-performing setting is highlighted in bold.

| Mask Ratio | MAR-B | MAR-L | MAR-H |
|---|---|---|---|
| 0.75 | 1.64 (-0.05) | 1.65 (-0.24) | 1.81 (-0.37) |
| 0.80 | 1.74 (+0.05) | 1.76 (-0.13) | 1.85 (-0.33) |
| 0.85 | 1.68 (-0.01) | 1.83 (-0.06) | 2.00 (-0.18) |
| 0.86 (default) | 1.69 (0.00) | 1.89 (0.00) | 2.18 (0.00) |
| 0.90 | 1.65 (-0.04) | 1.88 (-0.01) | 2.22 (+0.05) |
| **0.95** | **1.88 (+0.19)** | **2.25 (+0.36)** | **2.88 (+0.70)** |
| 0.99 | 1.77 (+0.08) | 1.86 (-0.03) | 2.14 (-0.04) |

**Fixed Timestep**   MIAs on DMs have been shown to be most effective when conducted at a specific denoising step $t$ (Carlini et al., 2023). Since MAR utilizes a small diffusion module for token generation, we hypothesize that targeting MIAs at a fixed timestep $t$ rather than a randomly chosen one can similarly enhance MIA effectiveness. Unlike full-scale diffusion models, where the most discriminative timestep is typically around $t = 100$, our experiments reveal that for MAR models, the optimal timestep is $t = 500$.

Table 12 illustrates the impact of this adjustment. When MIAs are performed at $t = 500$, MAR-H achieves a TPR@FPR=1% of 3.30, improving by +0.42 over the baseline random timestep approach. Similarly, MAR-L and MAR-B also see noticeable gains at this timestep. Notably, selecting timestep $t = 100$ significantly reduces the attack's effectiveness, with a drop of -0.38 for MAR-H.

Table 12: **Impact of using a fixed denoising timestep on MIAs for MAR performance.** We report TPR@FPR=1%. The most discriminative timestep is highlighted in bold.

| Timestep | MAR-B | MAR-L | MAR-H |
|---|---|---|---|
| random | 1.88 (0.00) | 2.25 (0.00) | 2.88 (0.00) |
| 100 | 1.60 (-0.27) | 1.90 (-0.34) | 2.50 (-0.38) |
| **500** | **1.88 (+0.00)** | **2.41 (+0.17)** | **3.30 (+0.42)** |
| 700 | 1.85 (-0.03) | 2.35 (+0.10) | 3.20 (+0.32) |
| 900 | 1.65 (-0.22) | 2.14 (-0.10) | 2.97 (+0.09) |

**Reducing Diffusion Noise Variance**    The MAR loss function, as defined in Equation (3), exhibits certain variance due to its dependence on randomly sampled noise $\epsilon$. During training, MAR uses four different noise samples per image. We hypothesize that increasing the number of noise samples can provide a more stable loss signal, thereby improving the performance of MIAs.

Our results, summarized in Table 13, confirm that increasing the number of noise samples has a positive effect on attack performance.

Table 13: **Impact of R reducing diffusion noise variance** on MIAs for MAR performance. We report TPR@FPR=1%. Obtaining loss for random noise sampled multiple times generally improves attack effectiveness. The best-performing setting is highlighted in bold.

| Repeats | MAR-B | MAR-L | MAR-H |
|---|---|---|---|
| 4 (default) | 1.88 (0.00) | 2.41 (0.00) | 3.30 (0.00) |
| 8 | 1.98 (+0.10) | 2.59 (+0.18) | 3.32 (+0.03) |
| 16 | 2.01 (+0.13) | 2.50 (+0.09) | 3.19 (-0.11) |
| 32 | 2.00 (+0.11) | 2.56 (+0.15) | 3.35 (+0.06) |
| **64** | **2.09 (+0.21)** | **2.61 (+0.20)** | **3.40 (+0.10)** |

# H. Full MIA Results

We report TPR@FPR=1% and AUC for each baseline MIA (Table 14, Table 15, each improved MIA for IAR (Table 16, Table 17) and each MIA for DMs (Table 18, Table 19). Results are randomized over 100 experiments.

Table 14: **TPR@FPR=1% for baseline MIAs.**

| Model | VAR-$d16$ | VAR-$d20$ | VAR-$d24$ | VAR-$d30$ | MAR-B | MAR-L | MAR-H | RAR-B | RAR-L | RAR-XL | RAR-XXL |
|---|---|---|---|---|---|---|---|---|---|---|---|
| Loss (Yeom et al., 2018) | 1.50±0.16 | 1.67±0.20 | 2.19±0.21 | 4.95±0.38 | 1.42±0.21 | 1.48±0.19 | 1.60±0.21 | 1.76±0.24 | 2.10±0.27 | 3.38±0.42 | 5.70±0.55 |
| Zlib (Carlini et al., 2021) | 1.55±0.20 | 1.74±0.20 | 2.24±0.24 | 5.77±0.59 | 1.41±0.22 | 1.49±0.21 | 1.59±0.22 | 1.91±0.23 | 2.45±0.26 | 4.21±0.31 | 7.52±0.57 |
| Hinge (Bertran et al., 2024) | 1.62±0.19 | 1.72±0.22 | 2.14±0.23 | 4.09±0.40 | — | — | — | 1.81±0.17 | 1.99±0.19 | 2.94±0.36 | 5.16±0.63 |
| Min-K% (Shi et al., 2024) | 1.58±0.16 | 2.04±0.25 | 3.22±0.38 | 12.23±1.13 | 1.69±0.18 | 1.89±0.16 | 2.18±0.23 | 2.09±0.24 | 2.86±0.32 | 5.83±0.52 | 13.48±0.98 |
| SURP (Zhang & Wu, 2024) | 1.53±0.17 | 1.70±0.20 | 2.23±0.23 | 5.02±0.43 | — | — | — | 1.84±0.18 | 2.12±0.30 | 3.46±0.46 | 5.82±0.53 |
| Min-K%++ (Zhang et al., 2024b) | 1.34±0.18 | 2.21±0.28 | 3.73±0.34 | 14.90±0.96 | — | — | — | 2.36±0.29 | 3.26±0.30 | 6.27±0.65 | 14.63±0.87 |
| CAMIA (Chang et al., 2024) | 1.33±0.18 | 1.76±0.19 | 3.07±0.35 | 16.69±1.16 | 1.35±0.19 | 1.38±0.19 | 1.44±0.23 | 1.51±0.17 | 1.78±0.15 | 1.99±0.34 | 4.34±0.51 |

Table 15: **AUC for baseline MIAs.**

| Model | VAR-$d16$ | VAR-$d20$ | VAR-$d24$ | VAR-$d30$ | MAR-B | MAR-L | MAR-H | RAR-B | RAR-L | RAR-XL | RAR-XXL |
|---|---|---|---|---|---|---|---|---|---|---|---|
| Loss (Yeom et al., 2018) | 52.35±0.35 | 54.53±0.34 | 59.55±0.35 | 75.45±0.30 | 51.92±0.36 | 53.33±0.36 | 55.06±0.34 | 54.92±0.37 | 58.04±0.37 | 65.59±0.34 | 74.45±0.30 |
| Zlib (Carlini et al., 2021) | 52.38±0.38 | 54.59±0.38 | 59.65±0.37 | 75.67±0.34 | 51.91±0.39 | 53.32±0.39 | 55.05±0.38 | 55.27±0.36 | 58.68±0.35 | 66.85±0.39 | 76.17±0.30 |
| Hinge (Bertran et al., 2024) | 53.29±0.39 | 56.83±0.39 | 62.89±0.39 | 77.36±0.33 | — | — | — | 57.07±0.44 | 61.41±0.44 | 71.48±0.39 | 82.14±0.29 |
| Min-K% (Shi et al., 2024) | 53.77±0.40 | 57.84±0.44 | 65.49±0.40 | 83.55±0.30 | 51.87±0.38 | 53.29±0.38 | 55.05±0.38 | 56.53±0.38 | 61.21±0.36 | 71.35±0.32 | 82.33±0.28 |
| SURP (Zhang & Wu, 2024) | 50.46±0.25 | 54.54±0.38 | 59.60±0.40 | 75.46±0.34 | — | — | — | 52.21±0.40 | 58.02±0.42 | 65.58±0.41 | 74.50±0.33 |
| Min-K%++ (Zhang et al., 2024b) | 54.52±0.41 | 57.93±0.38 | 65.76±0.38 | 85.33±0.27 | — | — | — | 57.82±0.41 | 62.48±0.38 | 75.61±0.32 | 85.16±0.26 |
| CAMIA (Chang et al., 2024) | 52.44±0.44 | 55.12±0.44 | 61.37±0.42 | 80.16±0.34 | 51.08±0.42 | 51.96±0.43 | 53.20±0.38 | 51.40±0.36 | 51.83±0.39 | 59.28±0.39 | 66.07±0.36 |

Table 16: **TPR@FPR=1% for our improved MIAs for IARs.**

| Model | VAR-$d16$ | VAR-$d20$ | VAR-$d24$ | VAR-$d30$ | MAR-B | MAR-L | MAR-H | RAR-B | RAR-L | RAR-XL | RAR-XXL |
|---|---|---|---|---|---|---|---|---|---|---|---|
| Loss (Yeom et al., 2018) | 2.16±0.26 | 5.95±0.54 | 24.03±1.91 | 86.38±0.92 | 1.54±0.22 | 1.81±0.21 | 2.26±0.26 | 2.86±0.20 | 5.50±0.39 | 16.58±0.97 | 40.76±1.87 |
| Zlib (Carlini et al., 2021) | 1.75±0.17 | 4.87±0.41 | 20.37±1.19 | 83.99±0.87 | 1.51±0.21 | 1.80±0.23 | 2.23±0.27 | 2.52±0.31 | 4.56±0.39 | 13.91±1.02 | 41.03±1.96 |
| Hinge (Bertran et al., 2024) | 0.00±0.00 | 0.00±0.00 | 0.00±0.00 | 0.00±0.00 | — | — | — | 2.50±0.20 | 4.34±0.39 | 10.59±0.88 | 20.23±1.85 |
| Min-K% (Shi et al., 2024) | 0.05±0.02 | 0.06±0.02 | 0.14±0.04 | 1.63±0.13 | 2.09±0.23 | 2.6±0.28 | 3.40±0.30 | 4.30±0.33 | 8.66±0.79 | 26.14±1.22 | 49.80±2.15 |
| Min-K%++ (Zhang et al., 2024b) | 0.39±0.06 | 1.40±0.11 | 4.88±0.20 | 37.90±0.44 | — | — | — | 4.19±0.40 | 8.24±0.66 | 23.04±1.14 | 43.67±2.32 |
| CAMIA (Chang et al., 2024) | 1.83±0.25 | 5.46±0.52 | 20.92±1.14 | 72.77±1.04 | 1.00±0.17 | 0.97±0.13 | 1.06±0.15 | 1.63±0.21 | 2.60±0.27 | 6.77±0.47 | 17.85±1.20 |

Table 17: **AUC for our improved MIAs for IARs.**

| Model | VAR-$d16$ | VAR-$d20$ | VAR-$d24$ | VAR-$d30$ | MAR-B | MAR-L | MAR-H | RAR-B | RAR-L | RAR-XL | RAR-XXL |
|---|---|---|---|---|---|---|---|---|---|---|---|
| Loss (Yeom et al., 2018) | 61.73±0.33 | 76.26±0.30 | 92.20±0.15 | 98.95±0.05 | 52.25±0.42 | 54.60±0.41 | 57.35±0.40 | 65.61±0.35 | 75.83±0.32 | 89.64±0.21 | 96.17±0.12 |
| Zlib (Carlini et al., 2021) | 57.91±0.39 | 70.86±0.33 | 88.69±0.24 | 98.51±0.07 | 52.23±0.39 | 54.57±0.39 | 57.33±0.39 | 62.22±0.42 | 72.19±0.37 | 87.51±0.22 | 95.46±0.13 |
| Hinge (Bertran et al., 2024) | 52.67±0.36 | 56.11±0.36 | 62.48±0.36 | 74.63±0.30 | — | — | — | 59.66±0.39 | 68.09±0.35 | 81.56±0.29 | 90.62±0.21 |
| Min-K% (Shi et al., 2024) | 59.78±0.34 | 70.43±0.34 | 83.10±0.25 | 90.16±0.27 | 53.31±0.40 | 56.34±0.39 | 59.98±0.38 | 66.81±0.38 | 78.73±0.32 | 91.36±0.20 | 96.97±0.10 |
| Min-K%++ (Zhang et al., 2024b) | 57.10±0.30 | 65.44±0.29 | 78.74±0.25 | 93.18±0.16 | — | — | — | 65.20±0.36 | 75.37±0.34 | 88.29±0.23 | 95.84±0.14 |
| CAMIA (Chang et al., 2024) | 56.37±0.38 | 68.18±0.31 | 84.83±0.24 | 96.95±0.09 | 50.86±0.41 | 51.15±0.41 | 51.75±0.41 | 57.95±0.40 | 63.17±0.43 | 70.43±0.39 | 83.55±0.31 |

Table 18: **TPR@FPR=1% of MIAs for DMs.**

| | LDM | U-ViT-H/2 | DiT-XL/2 | MDTv1-XL/2 | MDTv2-XL/2 | DiMR-XL/2R | DiMR-G/2R | SiT-XL/2 |
|---|---|---|---|---|---|---|---|---|
| Denoising Loss (Carlini et al., 2023) | 1.35±0.14 | 1.30±0.17 | 1.42±0.17 | 1.55±0.18 | 1.64±0.17 | 0.91±0.15 | 0.88±0.15 | 1.02±0.13 |
| SecMI$_{stat}$ (Duan et al., 2023c) | 1.30±0.20 | 1.31±0.19 | 1.49±0.22 | 1.35±0.17 | 1.52±0.22 | 1.15±0.21 | 1.05±0.15 | 0.00±0.00 |
| PIA (Kong et al., 2023) | 1.25±0.16 | 1.25±0.19 | 1.59±0.20 | 1.72±0.20 | 2.07±0.24 | 1.07±0.11 | 1.09±0.12 | 1.14±0.14 |
| PIAN (Kong et al., 2023) | 1.03±0.14 | 1.17±0.16 | 0.92±0.14 | 1.22±0.15 | 1.50±0.20 | 1.04±0.13 | 1.01±0.12 | 1.09±0.14 |
| GM (Dubiński et al., 2025) | 1.25±0.17 | 1.26±0.17 | 1.34±0.17 | 1.18±0.16 | 1.47±0.19 | 1.13±0.15 | 1.16±0.16 | 1.38±0.18 |
| ML (Dubiński et al., 2025) | 1.41±0.16 | 1.36±0.20 | 1.50±0.18 | 1.70±0.16 | 1.98±0.26 | 1.01±0.15 | 1.10±0.14 | 1.14±0.12 |
| CLiD (Zhai et al., 2024) | 1.55±0.19 | 1.75±0.22 | 2.08±0.28 | 2.72±0.39 | 4.91±0.44 | 0.96±0.14 | 0.90±0.13 | 6.38±0.64 |

Table 19: **AUC for MIAs for DMs.**

| | LDM | U-ViT-H/2 | DiT-XL/2 | MDTv1-XL/2 | MDTv2-XL/2 | DiMR-XL/2R | DiMR-G/2R | SiT-XL/2 |
|---|---|---|---|---|---|---|---|---|
| Denoising Loss (Carlini et al., 2023) | 50.53±0.41 | 50.36±0.42 | 51.77±0.43 | 51.25±0.37 | 51.65±0.37 | 46.25±0.40 | 46.01±0.40 | 47.25±0.34 |
| SecMI$_{stat}$ (Duan et al., 2023c) | 49.84±0.44 | 53.15±0.43 | 55.15±0.46 | 54.44±0.38 | 56.80±0.36 | 48.73±0.45 | 48.73±0.44 | 50.00±0.00 |
| PIA (Kong et al., 2023) | 48.97±0.43 | 51.77±0.44 | 53.18±0.42 | 52.60±0.44 | 54.68±0.45 | 47.31±0.42 | 47.16±0.41 | 49.13±0.44 |
| PIAN (Kong et al., 2023) | 49.56±0.43 | 50.99±0.46 | 50.14±0.43 | 49.96±0.42 | 51.52±0.38 | 49.85±0.41 | 49.79±0.43 | 50.17±0.37 |
| GM (Dubiński et al., 2025) | 51.51±0.40 | 51.19±0.42 | 50.46±0.46 | 50.72±0.39 | 48.85±0.37 | 45.97±0.45 | 45.86±0.45 | 50.94±0.38 |
| ML (Dubiński et al., 2025) | 50.36±0.41 | 51.16±0.41 | 52.53±0.45 | 50.42±0.19 | 54.65±0.38 | 46.26±0.38 | 49.37±0.41 | 49.83±0.17 |
| CLiD (Zhai et al., 2024) | 52.50±0.39 | 54.27±0.41 | 56.16±0.41 | 57.43±0.41 | 62.54±0.40 | 46.20±0.38 | 45.95±0.41 | 78.65±0.30 |

## I. Full DI Results

We report the outcome of DI for DMs in Table 20. As an additional observation, we note that contrary to DI for IARs, shifting from the classifier to an alternative feature aggregation increases the number of samples needed to reject $H_0$. This suggests, that the linear classifier remains necessary for DMs.

Table 20: **DI for DMs.** We report the minimal number of samples needed to successfully reject $H_0$.

| | LDM | U-ViT-H/2 | DiT-XL/2 | MDTv1-XL/2 | MDTv2-XL/2 | DiMR-XL/2R | DiMR-G/2R | SiT-XL/2 |
|---|---|---|---|---|---|---|---|---|
| DI for DM | 4000 | 700 | 400 | 300 | 200 | 2000 | 200 | 300 |
| No Classifier | 5000 | 4000 | 3000 | 600 | 400 | 2000 | 2000 | 500 |

# J. Mitigation Strategy

In this section we detail our privacy risk mitigation strategy.

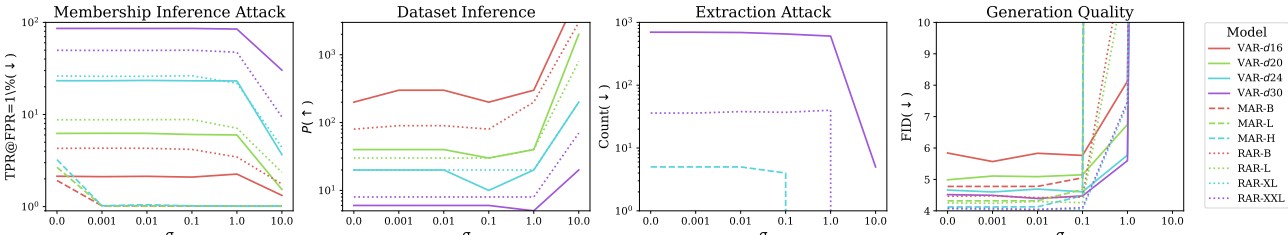

Figure 6: **Privacy-utility trade-off of our mitigation strategy.** We show that successfully defending VAR and RAR against MIA and DI requires adding noise that severely harms the performance. Interestingly, we are able to limit the extent of memorization for VAR, and fully defend MAR against MIA and DI.

## J.1. Method

Given an input sample $x$, we perturb the output of the IAR according to a noise scale $\sigma$, which we can adjust to balance privacy-utility trade-off. During inference, we add noise sampled from $\mathcal{N}(0, \sigma)$ to the output. For VAR and RAR, we add it to the logits, and for MAR we add them to the sampled continuous tokens.

We measure privacy leakage with our methods from Section 5. Specifically, we perform MIAs, DI, and the extraction attack. To quantify utility, we generate 10,000 images from the IARs, and compute FID (Heusel et al., 2017) between generations and the validation set. Lower FID means better quality of the generations.

## J.2. Results

Our results in Figure 6 show that we can effectively lower the privacy loss by applying our mitigation strategy, however, this comes at a cost of significantly decreased utility, as highlighted by substantially increasing FID score.

We are able to lower the MIAs success by more than half (Fig. 6, left), with the biggest relative drop observed for RAR-XL, for which the TPR@FPR=1% drops from 26% to **4.4%**. Moreover, *all* MAR models become immune to MIAs after noising their tokens, as TPR@FPR=1% drops to 1% (random guessing) with $\sigma = 0.001$. When we apply our defense to DI (Fig. 6, second from the left), we have to increase $P$, the minimum number required to perform a successful DI attack, by an order of magnitude, with the biggest relative difference for the smallest models: VAR-16, and RAR-B, with an increase from 80 to 3000, and 200 to 8000, respectively. Such an increase means that the models are harder to attack with DI, *i.e.,* their privacy protection is boosted. Similarly to MIA, DI stops working for MAR models immediately.

Our method achieves limited success in mitigating extraction (Fig. 6, third from the left). We are lowering the success of extraction attack only when adding significant amount of noise. However, for VAR-$d$30, which exhibits the biggest memorization, with $\sigma = 1.0$ we successfully protect **93** out of 698 samples from being extracted without significantly harming the utility. Our method, similarly to all defenses, suffers from lowered performance (Fig. 6, right), as signal-to-noise ratio during generation gets worse when $\sigma$ increases.

## J.3. Discussion

We show that we can mitigate privacy risks by adding noise to the outputs of IARs, at a cost of utility. Notably, all MARs become *fully* immune to MIAs and DI with noise scale as small as $0.001$. This result supports previous insights from Section 5, in which we show that MARs are significantly less prone to privacy risks than VARs and RARs. We argue that logits leak significantly more information than continuous tokens, and thus, adding noise to the latter yields significantly higher protection, at a lower performance cost.

We acknowledge that our privacy leakage defense is a heuristic, and more theoretically sound approaches should be explored, *e.g.,* in the domain of Differential Privacy (Dwork, 2006). To the best of our knowledge, we make the first step towards private IARs.

# K. More About Memorization

In this section we provide an extended analysis of memorization phenomenon in IARs. We show more examples of memorized images, highlight the relation between the prefix length $i$ and the number of extracted samples, and shed more light on our efficient extraction method, described in Section 5.3.

## K.1. More Memorized Images

In Figure 12 we show a non-cherry-picked set of images memorized by IARs. In Figure 7 we show an example of an image memorized verbatim by VAR-$d$30 **without any prefix**, *i.e.,* only from the class label token. In Figure 8 we show an image that has been memorized by both VAR-$d$30 and RAR-XXL.

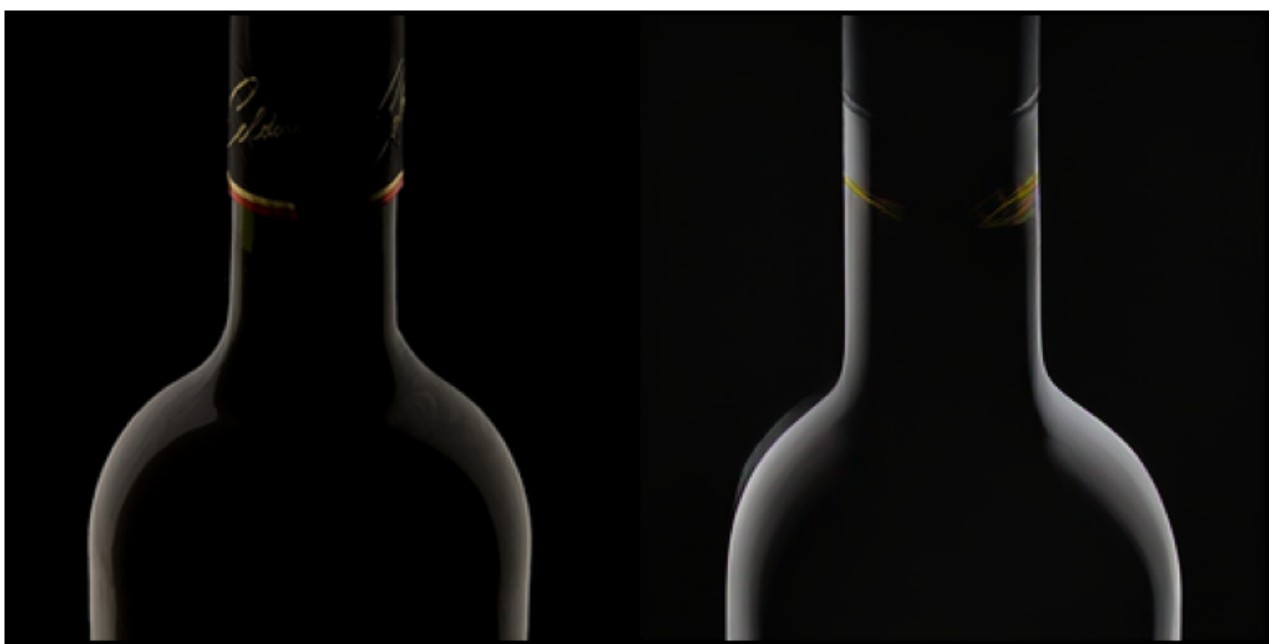

Figure 7: **Image extracted from VAR-$d$30 *without prefix*.** (Left) memorized image, (right) generated image.

## K.2. Prefix Length vs. Number of Extracted Images

We analyze the effect of the prefix length on the number of extracted samples. As our method leverages conditioning on a part of the input sequence, in Figure 10 we show an increase of extraction success with the increase in the length of the prefix. Notably, we start experiencing false-positives once the prefix length surpasses 30 for VAR-$d$30 and RAR-XXL, and 5 for MAR-H. In effect, the results in Table 4 provide an upper bound of the success of our extraction method.

Table 21: Prefix length $i$ for our data extraction attack. We note that appending longer sequences leads to false positives, *i.e.,* the IARs start to generate images from the validation set.

| Model | VAR-$d$30 | MAR-H | RAR-XXL |
|---|---|---|---|
| Prefix length $i$ | 30 | 5 | 30 |

## K.3. Approximate distance vs. SSCD Score

In this section we underscore the effectiveness of our filtering approach. Figure 11 shows that the distances we design for the candidate selection process indeed correlates with the SSCD score. By focusing only on the top-5 samples for each class we effectively narrow our search to just $0.5\%$ of the training set, significantly speeding up the whole process.

| Original | VAR-*d*30 | RAR-XXL |
|---|---|---|

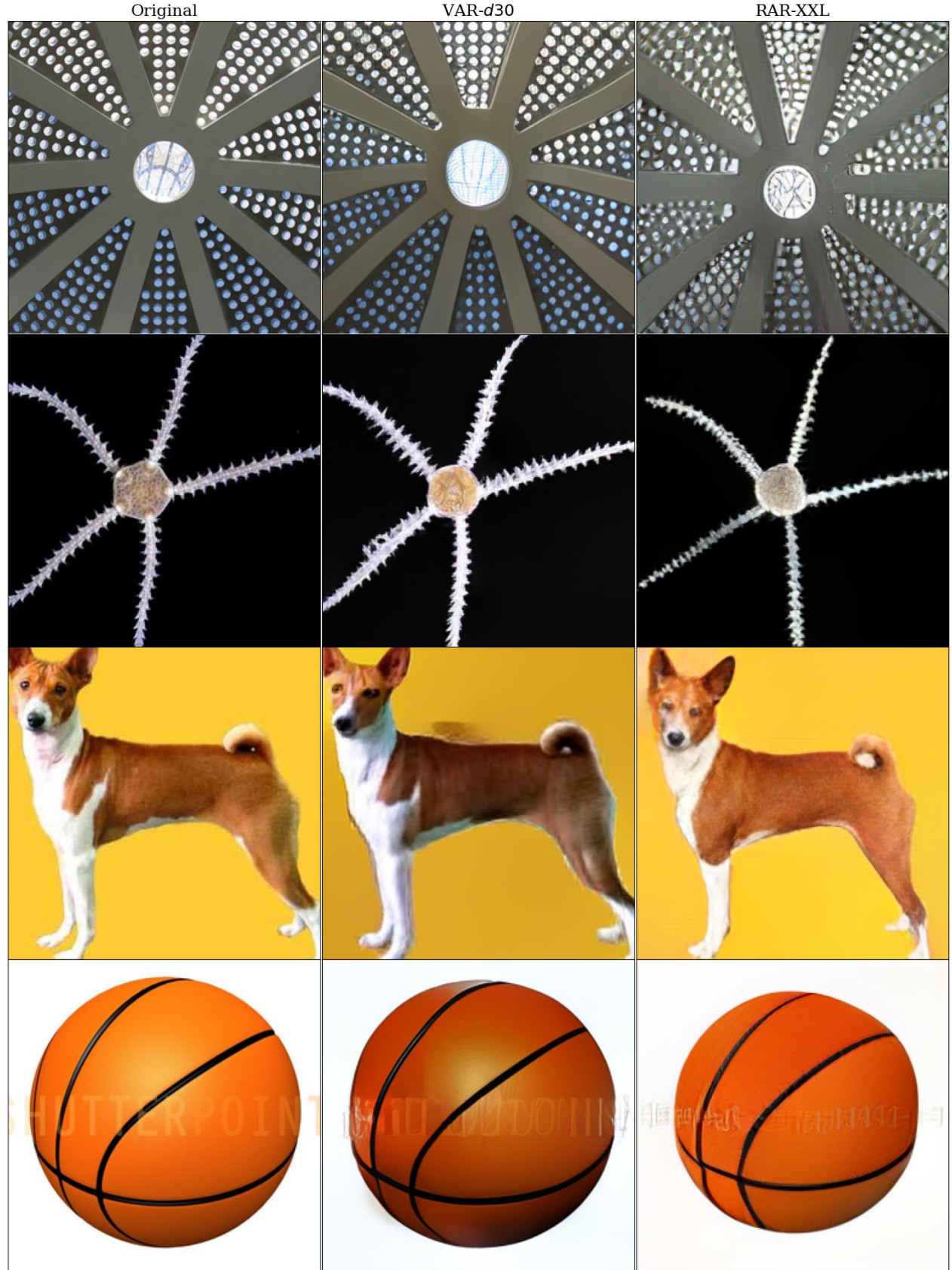

Figure 8: **Images extracted from both VAR-*d*30, and RAR-XXL.**

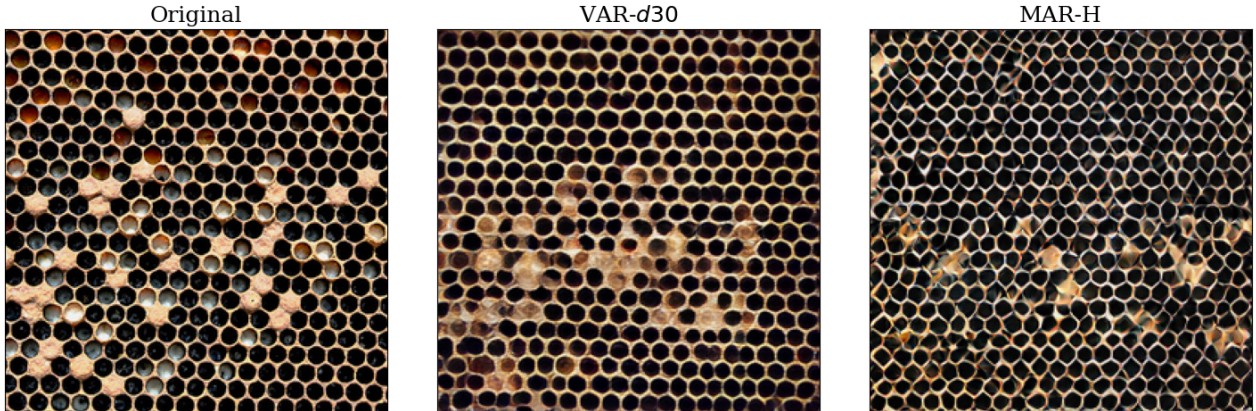

Figure 9: **An image extracted from both VAR-*d*30, and MAR-H.**

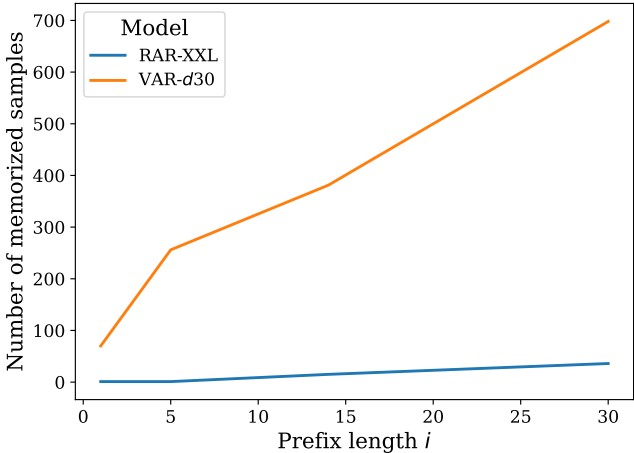

Figure 10: **Prefix length and the number of extracted samples.** We show that with an increase of the prefix length, the success of our extraction method increases.

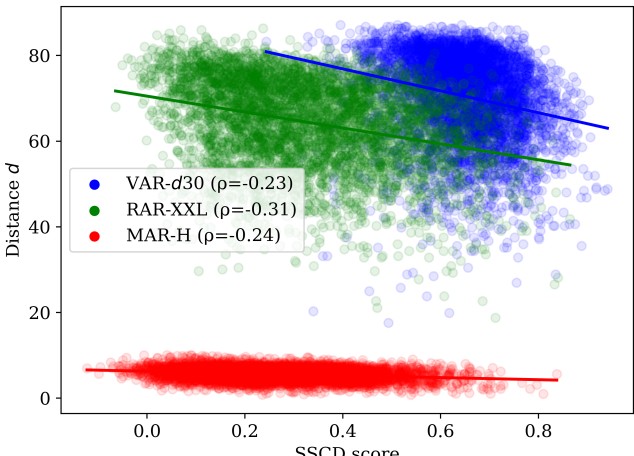

Figure 11: **Distance function *d* and the SSCD score.** We show that *d* correlates with the final memorization score. This result makes our candidate selection process sound, and reduces the cost of extracting memorized samples.

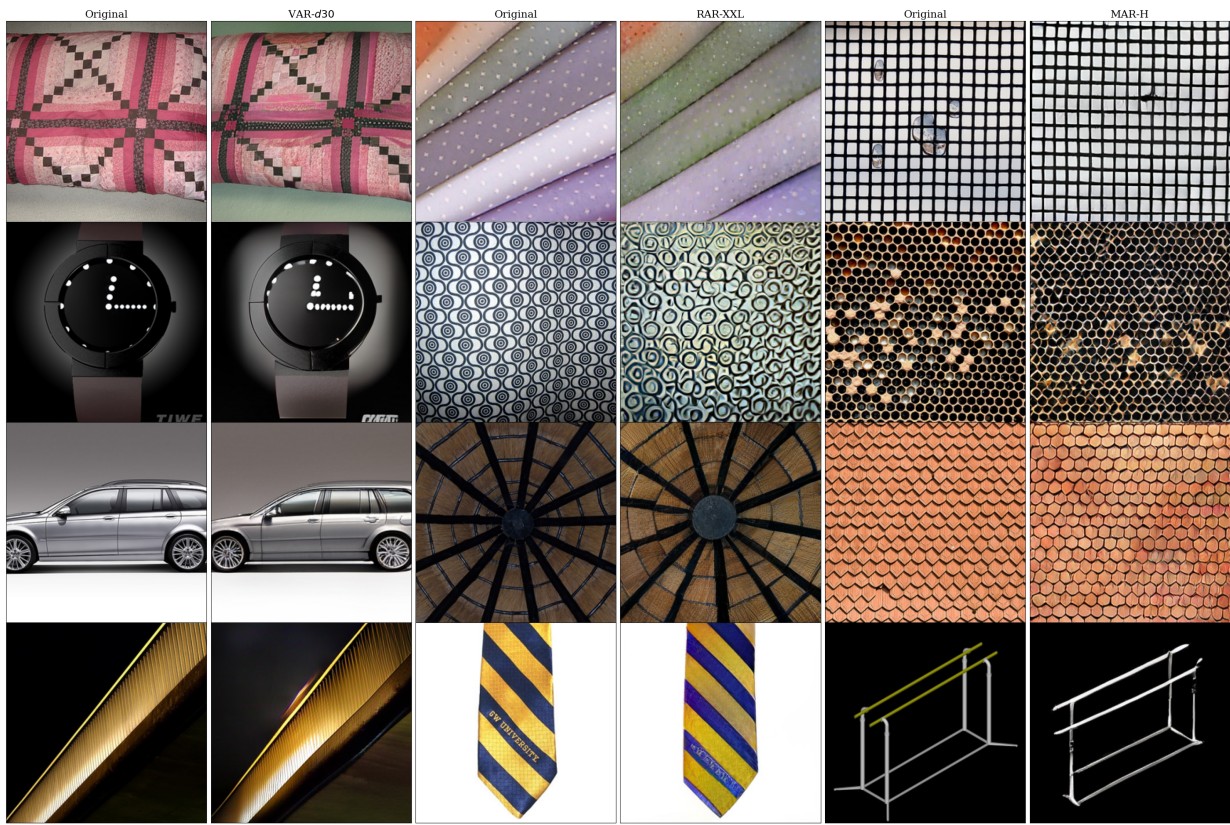

Figure 12: **Non-cherry-picked extracted images.** Odd columns from the left correspond to the original image, even to extracted. From left, the images are for VAR-*d*30, RAR-XXL, and MAR-H.

