# OpenReview forum: "Privacy Attacks on Image AutoRegressive Models"
_ICML.cc/2025/Conference — ICML 2025 poster_

### Official Review · Reviewer_1YbK · 2025-02-25

**Overall Recommendation:** 2

**Summary:**

The paper presents a systematic analysis of privacy attacks on image autoregressive models, including membership inference attacks, dataset inference, and data extraction attacks. The proposed method is primarily constructed from components of previous work.

## Update after Rebuttal

The authors have provided a detailed response and addressed most of my concerns regarding the experimental aspects. However, my main concern remains the clarity of the writing. While it is encouraging that the authors have outlined a plan for revision, it is difficult to fully assess the impact of these changes without seeing the modified version of the paper. (Although the authors note that updates are not allowed by the conference, the current version is indeed unclear.) Therefore, I still lean toward a weak reject and recommend that the paper be revised and resubmitted to another venue.

**Claims And Evidence:**

In the abstract, the third contribution states:
> IARs outperform DMs in generation efficiency and quality but suffer order-of-magnitude higher privacy leakage compared to them in MIAs, DI, and data extraction.

However, since the authors propose a privacy-related attack specifically tailored for IARs, it is misleading to claim that IARs inherently have higher privacy leakage. It would be more accurate to state that *using the tailored attack*, higher privacy leakage is observed in IARs compared to DMs. Privacy leakage measurements should reflect an upper bound across possible methods, not just the results from a specific, targeted attack.

**Essential References Not Discussed:**

Not found.

**Experimental Designs Or Analyses:**

Experiment design is mostly good.

**Methods And Evaluation Criteria:**

I think the method is mostly intuitive but lacks sufficient emphasis on the differences from previous work. The paper’s writing style tends to merge several aspects into its contributions, which should be clarified. For example:

1. In the introduction:
> We exploit this property and compute the difference in outputs between conditional and unconditional inputs as an input to MIAs.

At minimum, a citation to CLiD should be included here. Without it, this technique appears to be the author’s own contribution.

2. In Section 5.3:
> This attack builds on elements of data extraction attacks for LLMs (Carlini et al., 2021) and DMs (Carlini et al., 2023).

However, it is not clearly stated which parts align with previous work. For instance, fixing and knowing the first *i* tokens directly mirrors the setting in LLMs (Carlini et al., 2021).

I believe the method includes some novel designs, but the unclear presentation of prior work makes the paper’s contributions less apparent.

**Other Comments Or Suggestions:**

Although the attacking method is not particularly innovative compared to previous work, the paper appears solid due to its strong experimental section. However, the unclear articulation of the contributions gives the impression of claiming a larger contribution than is warranted.

**Other Strengths And Weaknesses:**

The experimental section is solid, but the proposed method appears somewhat empirical. It would be beneficial to include more insights. For instance, the approach in Section 5.1 feels like parameter tuning, focusing on the timestep and binary mask ratio, rather than offering deeper methodological innovations.

**Questions For Authors:**

See comments above.

**Relation To Broader Scientific Literature:**

There has been extensive research on privacy leakage threats in LLMs and DMs. IARs, which can be seen as a new architecture combining properties of both LLMs and DMs, have not been explored in the context of privacy leakage. This paper addresses that gap.

**Theoretical Claims:**

No theoretical claims found.

---

> ### Author Rebuttal · Authors · 2025-04-01
>
> >**The method is primarily constructed from components of prev. work.**
>
> Beyond the proposed method, our contributions are:
>
> 1. **First empirical privacy leakage evaluation of IARs**. We develop the strongest model-specific attacks, and perform comprehensive analysis across publicly available models.
>
> 2. **Privacy-utility trade-off**. We show IARs are fast and performant, but substantially less private. We highlight that DMs are comparably performant, while leaking *significantly less* information about the training data.
>
> >**The authors propose an attack specifically tailored for IARs, it is misleading to claim that IARs inherently have higher privacy leakage. It would be more accurate to state that *using the tailored attack*, higher privacy leakage is observed in IARs compared to DMs. Privacy leakage measurements should reflect an upper bound across possible methods, not just the results from a specific, targeted attack.**
>
> We do not use IAR-tailored MIAs/DI against DMs–they are not applicable to DMs. **We use SOTA DM-specific attacks against DMs**, thus the observed privacy leakage for DMs and IARs **is an empirical upper bound** across possible methods, since we use the strongest possible attacks. Effectively, our claims hold. We improved the wording in the manuscript.
>
> >**The method [...] lacks sufficient emphasis on the differences from previous work. [...]  In the introduction: [...] a citation to CLiD should be included.**
>
> We now include the citation to CLiD (Zhai et al. 2024) in the introduction.
>
> Our MIA for VAR/RAR provides the following innovations over CLiD:
>
> 1. **Difference between logits, not model loss**. CLiD uses $\mathcal{L}(x,c)-\mathcal{L}(x,c_{null})$. Since MIAs we use work with logits, we compute $p(x|c)-p(x|c_{null})$.
>
> 2. **Parameter-free method**. CLiD needs a sweep through a hyperparameter $\alpha$ to achieve its high performance, as well as Robust-Scaler to stabilize the MIA signal. We provide a more generalized approach.
>
> >**2. In Sec. 5.3: “[...] builds on elements of data extraction attacks [...].” It is not clearly stated which parts align with previous work. [...] Fixing and knowing the first i tokens mirrors the setting in LLMs.**
>
> Our attack consists of the following:
> 1. **Efficient candidates selection**. We do not simply generate millions of images. Instead, we identify promising images for further generation. This improves over Carlini et al., 2023.
>
> 2. **Fixing the prefix**. We directly follow the approach by Carlini et al., 2021.
>
> 3. **Generating from the prefix**. We follow (Carlini et al., 2021), and use greedy sampling for VAR/RAR starting from the prefix. For MAR, we do not alter the generation process.
>
> 4. **Final assessment**. In contrast to LLM extraction (Carlini et al., 2021), we focus on the images, not sequences of tokens. Samples we extract *do not match* the training samples in the token space; instead, we classify a sample as extracted in the image domain. To this end, we use SSCD (Pizzi et al., 2022), following Wen et al., 2024.
>
> >**[...] the unclear presentation of prior work makes the paper’s contributions less apparent. [...] the attacking method is not particularly innovative [...].**
>
> Our methods include novel designs:
> 1. **Improved MIA against VAR/RAR**. While we *are* building on CLiD, the adaptation to IARs is non-trivial. We improve the performance by up to **69%**.
>
> 2. **MIA specifically tailored against MAR**. We exploit model vulnerabilities (the DM module, training specifics) to boost the performance of our attack.
>
> 3. **Efficient data extraction**. While we build on previous work, we address two drawbacks: 1) focus in the sequence space, 2) high cost. We suggest an improved method enabling quicker, more successful extraction from the IARs, extracting up to **698** images.
>
> 4. **Improved DI**. We improve feature aggregation (replacing scoring function) and the feature extraction pipeline. We achieve an improvement of up to over **90%** compared to the baseline.
>
>
> >**The experimental section is solid, but [...] it would be beneficial to include more insights.**
>
> We appreciate that the Reviewer considers our experiments solid. We agree that insights would improve our work. In the [answer to the Reviewer RYDX](https://openreview.net/forum?id=7SXXczJCWP&noteId=GeVeD4wCQI) we provide intuitions to *why* IARs are more vulnerable than DMs. We add them to the paper.
>
> >**The approach in Sec. 5.1 feels like parameter tuning [...] rather than offering deeper methodological innovations.**
>
> Our main innovation stems from modifying the DM module of MAR to increase the leakage. We are not aware of any prior work that modifies the inference stage of an AR model to design a more potent attack.
>
> >**[...] unclear articulation of the contributions gives the impression of claiming a larger contribution than is warranted.**
>
> Do our above answers articulate our contributions clearly enough? We are happy to incorporate any additional feedback.

---

> > ### Comment · Reviewer_1YbK · 2025-04-03
> >
> > I feel the authors may not have fully understood my point. To clarify, I’m not suggesting that the authors are using *IAR-tailored* MIAs/DI attacks against DMs. Rather, my concern is with the strength of the claim being made:
> > > *"IARs outperform DMs in generation efficiency and quality but suffer order-of-magnitude higher privacy leakage compared to them in MIAs, DI, and data extraction."*
> >
> > Making such a statement implies that one model structure inherently poses a greater privacy risk than another. To support a claim of this nature, it’s important to use a standardized attack that is equally applicable across all model types, rather than relying on attacks specifically tailored to one kind of model. Otherwise, if a newly developed, tailored MIA for DMs were to achieve higher accuracy in the future, would that then suggest DMs have more privacy leakage than IARs? We all understand that attack methods are constantly evolving and improving. Given that the paper positions structural comparison as a key contribution, I see this as a significant concern.
> >
> > As for the rest of the rebuttal, I appreciate the additional explanations and clarifications. However, the extent of the changes—especially in writing and framing—is substantial. I believe these modifications should be fully incorporated into the paper itself. Until I see a revised version, I don’t feel comfortable adjusting my score.

---

> > > ### Author Response · Authors · 2025-04-04
> > >
> > > >**I feel the authors may not have fully understood my point. To clarify, I’m not suggesting that the authors are using IAR-tailored MIAs/DI attacks against DMs. Rather, my concern is with the strength of the claim being made: "IARs outperform DMs in generation efficiency and quality but suffer order-of-magnitude higher privacy leakage compared to them in MIAs, DI, and data extraction." Making such a statement implies that one model structure inherently poses a greater privacy risk than another. To support a claim of this nature, it’s important to use a standardized attack that is equally applicable across all model types, rather than relying on attacks specifically tailored to one kind of model. Otherwise, if a newly developed, tailored MIA for DMs were to achieve higher accuracy in the future, would that then suggest DMs have more privacy leakage than IARs? We all understand that attack methods are constantly evolving and improving. Given that the paper positions structural comparison as a key contribution, I see this as a significant concern.**
> > >
> > > We greatly appreciate the Reviewer’s clarification of their point. We find the Reviewer’s concerns sound and valid.
> > >
> > > We are happy to provide evaluation of the privacy risks for DMs and IARs under a *unified attack* for all models. To this end, we employ *Loss Attack* (Yeom et al., 2018), which uses model loss as the input, and is model-agnostic. For DMs we compute MSE between noise prediction and input noise (model loss, Equation 3 in the paper) at a **random timestep** (instead of a fixed t=100 following [1]) and for a **single noise** (instead of 5 following [1]). For MAR we discard all the improvements to the MIAs (fixed timestep, multiple noises, optimal mask ratio), and compute the mean of per token loss (Equation 3 in the paper). For VAR and RAR we compute the mean of per-token Cross-Entropy loss (Equation 2).
> > >
> > > We also unify the DI attack: we remove the scoring function $s$ for both IARs and DMs, and run the t-test on the single feature–Loss Attack’s output. The results for all models are below:
> > >
> > >
> > > |Model|Architecture|$P$ (DatasetInference)|TPR@FPR=1% (MIA)|AUC (MIA)|Accuracy (MIA)
> > > -|-|-|-|-|-
> > > VAR-d16|IAR|3000|1.50|52.35|50.08
> > > VAR-d20|IAR|1000|1.67|54.54|50.11
> > > VAR-d24|IAR|300|2.19|59.56|50.15
> > > VAR-d30|IAR|40|4.95|75.46|50.32
> > > MAR-B|IAR|6000|1.43|51.31|50.48
> > > MAR-L|IAR|3000|1.52|52.35|50.70
> > > MAR-H|IAR|2000|1.61|53.66|51.07
> > > RAR-B|IAR|800|1.77|54.92|50.25
> > > RAR-L|IAR|400|2.10|58.03|50.39
> > > RAR-XL|IAR|80|3.40|65.58|50.81
> > > RAR-XXL|IAR|40|5.73|74.44|51.64
> > > |
> > > LDM|DM|>20000|1.08|50.13|50.13
> > > U-ViT-H/2|DM|>20000|0.85|50.11|50.07
> > > DiT-XL/2|DM|>20000|0.84|50.09|50.15
> > > MDTv1-XL/2|DM|>20000|0.85|50.05|50.08
> > > MDTv2-XL/2|DM|>20000|0.87|50.14|50.16
> > > DiMR-XL/2R|DM|>20000|0.89|49.55|49.70
> > > DiMR-G/2R|DM|>20000|0.85|49.54|49.69
> > > SiT-XL/2|DM|6000|0.95|48.22|49.97
> > >
> > > Our results for the unified attack are consistent with the other results (Tables 1, 3, 13). Empirical data shows that IARs are more vulnerable to MIAs and DI. Loss Attack does not yield TPR@FPR=1% greater than random guessing (1%) for DMs, whereas all IARs perform above random guessing. Moreover, with such a weak signal, DI ceases to be successful for DMs, requiring above 20,000 samples ($P$) to reject the null hypothesis (no significant difference between members and non-members), with one exception: SiT. Conversely, IARs retain their high vulnerability to DI, with the most private IAR--MAR-B–being similarly vulnerable to the least private DM--SiT.
> > >
> > > We believe results obtained under the unified attack strengthen our message that current IARs leak more privacy than DMs.
> > >
> > > >**As for the rest of the rebuttal, I appreciate the additional explanations and clarifications. However, the extent of the changes—especially in writing and framing—is substantial. I believe these modifications should be fully incorporated into the paper itself. Until I see a revised version, I don’t feel comfortable adjusting my score.**
> > >
> > > We are happy to submit the updated paper (we inquired the AC regarding this option, as neither updating the submission nor providing link to extra text seems to be allowed this edition, via https://icml.cc/Conferences/2025/PeerReviewFAQ#discussions).
> > > In the current response, we have included a detailed breakdown of the changes made to the manuscript, along with corresponding line numbers:
> > >
> > > 1. We highlight the differences between our MIA against VAR/RAR and CLiD (lines 235-245).
> > >
> > > 2. We improved the presentation of our contribution–a more efficient data extraction method (lines 411-417).
> > >
> > > 3. We included a section with thorough discussion about the inherent properties of IARs that increase the leakage compared to DMs (lines 327, 328, 374, 375, Appendix).
> > >
> > > 4. We added the setup, results, and explanation of the unified attack suggested by the Reviewer (Appendix).
> > >
> > > The Camera Ready version will incorporate the above changes if accepted.
> > >
> > > We thank the Reviewer for the valuable feedback and hope that our answers address all the concerns.

---

### Official Review · Reviewer_rfqr · 2025-03-10

**Overall Recommendation:** 3

**Summary:**

This paper provides a thorough investigation into the privacy risks of image autoregressive models (IARs), highlighting their elevated vulnerability compared to diffusion models (DMs). The authors develop a novel membership inference attack (MIA) with significantly higher detection rates, introduce a dataset inference (DI) method requiring notably fewer samples, and demonstrate large-scale data extraction from IARs. Moreover, the authors conclude that a critical privacy-utility trade-off: while IARs outperform DMs in terms of image generation quality and speed, they are more susceptible to privacy breaches.



## Update After Rebuttal
The author has addressed my initial concerns, particularly on baseline comparisons and experimental settings. However, issues with wording and narration persist, which could cause misunderstandings. I recommend addressing these if the paper is accepted.
I also agree with Reviewer 1YbK's concerns about the conclusion that "IARs have higher privacy leakage." The attacks are tailored for IARs, so a more cautious phrasing like "empirical upper bound" or "tend to" would be more appropriate.
Overall, the experimental section is robust and provides insights for other researchers, I lean toward a weak accept.

**Claims And Evidence:**

1. The paper claims that autoregressive image models (IARs) are inherently more vulnerable to privacy attacks than diffusion models (DMs). However, the proposed membership inference and dataset inference attacks are specifically tailored to exploit autoregressive architectures, raising questions about whether the comparisons with DMs are conducted under balanced conditions.

2. It asserts that the proposed membership inference attacks significantly improve performance on AR models (e.g., a TPR of 86.38% at 1% FPR and up to 69% improvement over previous methods). However, different MIA strategies are tailored for MAR and for VAR/RAR, prompting the question of whether a single, unified MIA strategy should be used to conclusively evaluate performance.

**Essential References Not Discussed:**

The current reference are sufficient.

**Experimental Designs Or Analyses:**

1. The experiments are structured to compare privacy vulnerabilities between IARs and DMs, with a focus on the performance of tailored membership inference attacks. However, the experimental setup may favor IARs by using attack strategies specifically optimized for them, while applying different (and potentially less aligned) approaches for MAR and VAR/RAR. This design choice raises concerns about whether the performance differences are due to inherent model vulnerabilities or inconsistencies in the attack strategies applied.

2. The authors select certain baselines from language model research, but some vital experimental settings are not explicitly detailed. For instance, the term “baseline” appears repeatedly in Tables 1, 2, and 3 without fully describing the precise settings, assumptions, and hyperparameters employed.

**Methods And Evaluation Criteria:**

1. The study employs tailored membership inference attacks specifically designed for autoregressive architectures. However, the rationale behind selecting certain baselines—often derived from language model research—raises questions about their suitability for diffusion models. Additionally, clarity in methodological descriptions (e.g., the integration of CLiD into MIAs) and the explicit definition of experimental settings (such as hyperparameters) require improvement.

2. Evaluation is based on performance metrics such as TPR at fixed FPR and comparative improvements over baseline methods. The evaluation metrics themselves appear to be sound and reasonable.

**Other Comments Or Suggestions:**

1. For Figures 1 and 2, it is recommended to avoid using dashed lines ('----') for interval division, as they might be mistaken for elements of the legend. Instead, consider incorporating these distinctions directly within separate legend entries. Additionally, employing triangle symbols for diffusion-based methods could enhance visual differentiation.
2. Including a full extension or definition of abbreviation TPR@FPR in the abstract or introduction would help readers unfamiliar with the metric to understand its significance and context.

**Other Strengths And Weaknesses:**

Strengths:

1. The paper offers a well-structured background section that thoroughly contextualizes privacy attacks on generative models. This makes the work accessible to researchers who are new to privacy attack techniques yet familiar with generative modeling.

2. This study appears to be the first systematic evaluation of privacy attacks specifically targeting autoregressive image models, which have gained prominence due to their speed and quality benefits. The direct comparison with diffusion-based models adds valuable insights to the research community.

3. The proposed membership inference attacks show remarkable enhancements over naive baselines. Notably, a TPR of 86.38% at 1% FPR and improvements up to 69% over previous methods underscore the effectiveness of the tailored attacks.

Weaknesses:

1. While the paper concludes that IARs, with their superior generation speed and image quality, are more susceptible to privacy breaches than DMs, the direct comparisons may not be entirely equitable. The proposed membership inference and dataset inference attacks are specifically tailored to exploit autoregressive architectures, raising questions about whether the comparisons with DMs are conducted under balanced conditions. To strengthen the argument that IARs inherently pose a higher privacy risk, the authors might consider either developing equally specialized attacks for diffusion models or applying the same, more generic attack methods to both model types. This would help ensure that any observed differences in vulnerability stem from the model architectures themselves rather than from a mismatch in attack strategies.

2. The paper aims to demonstrate two core points: (1) IARs are intrinsically more vulnerable to privacy attacks than DMs, and (2) the newly proposed attacks outperform existing methods. However, the current experimental setup appears to favor IARs as a more accessible target from the outset. In particular, the chosen baselines—many of which originate from language model research—may not offer the most relevant or rigorous benchmarks for diffusion-based image models. To reinforce the claim that IARs are inherently more susceptible, the authors should clarify why LLM-attack approaches were selected over methods designed for DMs-attack, and detail how these baselines align or diverge from DM-specific attacks. A clearer justification of baseline choices, as well as a more balanced experimental design, would further bolster the credibility of the results and conclusions.

3. Certain technical implementations lack detailed descriptions, making reproducibility challenging. For instance, the authors briefly state that they “incorporate [this] into our MIAs by building on CLiD” (lines 222–224), but do not elaborate on how this integration is achieved. Similarly, the description of how the tailored MIA approach is employed within the DI framework (lines 315–318) remains vague, limiting the clarity of the specific methodological contributions.

4. Although the paper highlights in Section 5.1 that different MIA strategies are tailored for MAR and for VAR/RAR, these strategies are combined into a single table (Table 1) without clearly distinguishing how each approach is evaluated. This makes it difficult to discern whether the metrics for MAR should be compared directly to those for VAR/RAR, especially if they rely on different methodologies.
Additionally, the term “baseline” appears repeatedly in Tables 1, 2, and 3, but the precise settings, assumptions, and hyperparameters for these baselines are not fully described. This lack of clarity complicates the interpretation of experimental results and raises questions about whether comparisons across methods are valid. To improve transparency and rigor, the authors should clearly demarcate the distinct MIA strategies (e.g., MAR vs. VAR/RAR) and provide detailed descriptions of the baselines, including all relevant configurations and parameter choices.

**Questions For Authors:**

1. Please explain why the membership inference and dataset inference attacks are specifically tailored to exploit autoregressive architectures to conclude that IARs are more susceptible to privacy breaches than DMs. This design choice seems to favor ARs and may impact the fairness of the comparison.

2. Please elaborate on the unique contributions or key design elements of your methods. The paper currently lacks a clear theoretical contribution beyond statements such as “incorporate CLiD in our methods” (line 224) and “simply summing” (line 289). A more detailed explanation of the underlying principles would strengthen the work.

3. Please provide detailed descriptions of the experimental settings. Specifically, clarify the precise configurations, assumptions, and hyperparameter choices for the baselines mentioned in Tables 1, 2, and 3, which are currently seem to be described in a fuzzy manner.

**Relation To Broader Scientific Literature:**

N/A

**Theoretical Claims:**

No, the paper does not present explicit theoretical proofs for the proposed methods. Instead, it primarily relies on the intrinsic design of the approaches and empirical validation through experimental results. While the work may be conceptually motivated, it does not include rigorous theoretical justifications or formal proofs.

---

> ### Author Rebuttal · Authors · 2025-04-01
>
> We thank the Reviewer for the feedback.
>
> ## Attacks tailored for LLMs, DMs, IARs
>
> >**The proposed [MIA and DI] are specifically tailored to exploit [ARs], raising questions about whether the comparisons with DMs are conducted under balanced conditions. [Selecting baselines derived] from language model research raises questions about their suitability for [DMs]. [...] The experimental setup appears to favor IARs. [...] The baselines [...] may not [be] rigorous benchmarks for [DMs].**
>
> We use MIAs that are tailored to unique characteristics of a given model, including DMs (see Tab. 13, App. F). For DMs we use **the strongest** MIA available at the time of writing the paper, namely CLiD (Zhai et al., 2024), and we *do not* use LLM/IAR-specific MIAs against DMs. Similarly, for IARs we build on the strongest MIAs that are suitable for the AR architecture of these models. Overall, we use the strongest attacks for a given model, following [1].
>
> >**[...] developing equally specialized attacks for [DMs] or applying the same, more generic attack methods [...]. The authors should clarify why LLM-attack approaches were selected over methods designed for DMs.**
>
> We *do not* use LLM/IAR-specific MIAs/DI against DMs. Instead, we use SOTA DM-specific attacks, as explained in the previous answer.
>
> >**Why [MIAs and DI] tailored [for ARs show that] IARs are more susceptible to privacy breaches than DMs. This design choice seems to favor ARs.**
>
> To perform DI against DMs, we use CDI (Dubiński et al., 2024)–a method *explicitly* created for DMs. For IARs we build upon LLM DI (Maini et al., 2024), which we adapt to IAR specifics (see Sec. 5.2) to ensure a fair comparison.
>
> >**Different MIA strategies are tailored for MAR and for VAR/RAR [...] (Should?) a single, unified MIA strategy be used [...]**
>
> Empirical privacy leakage analysis should be carried out with respect to the worst case [1], thus the strongest known attack. Unified MIA for all IARs would not allow such comparison.
>
> >**[...] whether the performance differences are due to inherent model vulnerabilities or inconsistencies in the attack strategies [...] These strategies are combined into a single tab. without clearly distinguishing how each approach is evaluated. [It is unclear if] the metrics for MAR should be compared directly to those for VAR/RAR.**
>
> MAR and V/RAR are distinct in *design, inference, and training–and the attacks differ too*, as they exploit unique vulnerabilities of the models. Some design choices allow for stronger attacks; it is reflected in the results (Tab. 1). Evaluation protocol stays consistent; the attacks vary.
>
> ## Baselines
>
> >**The authors select certain baselines [...], but some vital experimental settings are not explicitly detailed. For instance, the term “baseline” appears repeatedly in [Tab. 1-3] without fully describing the precise settings, assumptions, and hyperparameters [...]**
>
> All MIAs assume gray-box access to the model, i.e., output and model loss. Some MIAs for DMs require white-box access to the model. We expanded the description provided in App. B. For all MIAs, we use the default hyperparameters from the respective MIAs. Following the literature we report the TPR@FPR=1% only for the best hyperparameter in Table 9, 11. In Table 1, for baseline and our methods, we report the best MIA for each model, as we strive to compare only *the strongest* attacks.
>
> ​​In Tab. 1-3, “baseline” denotes a naive use of LLM-tailored MIAs and DI to attack IARs. We revise App. B to include experimental details, relevant configurations and parameter choices.
>
> >**A clearer justification of baseline [...]**
>
> For IARs we use MIAs and DI for LLMs as “Baseline” (Table 1-3) as we can directly apply them to IARs, and no IAR-specific attacks exist in prev. work.
>
> ## Other
>
> >**Methodological descriptions (e.g., the integration of CLiD into MIAs) and the explicit definition of experimental settings [...] require improvement. [For instance] “incorporate [this] into our MIAs by building on CLiD”, [...] how the tailored MIA approach is employed within the DI [...].**
>
> We improved the clarity of those aspects. For details, see answer to the [Reviewer ZL49](https://openreview.net/forum?id=7SXXczJCWP&noteId=yF02U11EAL]).
>
> >**Fig. 1 and 2**
>
> We improve Fig. 1,2 accordingly.
>
> >**TPR@FPR**
>
> We clarify what TPR@FPR (True Positive Rate at False Positive Rate) stands for in the abstract section.
>
> >**Please elaborate on the unique contributions or key design elements of your methods. The paper lacks a clear theoretical contribution beyond statements such as “incorporate CLiD in our methods” (line 224) and “simply summing” (line 289).**
>
> Please refer to [the answer to the Rev. 1YbK.](https://openreview.net/forum?id=7SXXczJCWP&noteId=yJHAnkGBJf)
>
> >**Tab. 9**
>
> We thank the Reviewer for the suggestion. We will move Tab. 9 to the main text if accepted, given the 1 extra page allowed.
>
> **Ref.:**
>
> [1] Carlini et al., Extracting Training Data from [DMs], USENIX 2023.

---

> > ### Comment · Reviewer_rfqr · 2025-04-02
> >
> > The author has made progress in addressing my initial concerns, particularly regarding fairness in baseline comparisons, experimental settings, and the manuscript's contributions to the field. However, I believe the manuscript still lacks in wording and narration, which could potentially lead to misunderstandings. I encourage the author to address these concerns should the paper be accepted.
> >
> > Furthermore, I share Reviewer 1YbK’s concerns regarding the conclusion that "IARs have higher privacy leakage." I find this assertion problematic, as the attacks discussed in the paper are specifically tailored for IARs. A more cautious phrasing, such as "empirical upper bound" or "tend to," would be more appropriate than the definitive statement "our comprehensive analysis demonstrates that IARs exhibit significantly higher privacy risks than DMs."
> >
> > However, the experimental section of the paper is robust and provides some insights for researchers. Consequently, I am inclined to give a weak accept overall.

---

> > > ### Author Response · Authors · 2025-04-02
> > >
> > > >**The author has made progress in addressing my initial concerns, particularly regarding fairness in baseline comparisons, experimental settings, and the manuscript's contributions to the field. However, I believe the manuscript still lacks in wording and narration, which could potentially lead to misunderstandings. I encourage the author to address these concerns should the paper be accepted.**
> > >
> > > We thank the Reviewer for the valuable feedback. The manuscript has greatly improved by incorporating it. We will revise the Camera Ready manuscript should the paper be accepted to ensure maximum clarity.
> > >
> > >
> > > >**Furthermore, I share Reviewer 1YbK’s concerns regarding the conclusion that "IARs have higher privacy leakage." I find this assertion problematic, as the attacks discussed in the paper are specifically tailored for IARs. A more cautious phrasing, such as "empirical upper bound" or "tend to," would be more appropriate than the definitive statement "our comprehensive analysis demonstrates that IARs exhibit significantly higher privacy risks than DMs."**
> > >
> > > In the revised manuscript, we temper our claims to be more precise and state that we find the privacy risks for IARs are *empirically* more severe than the ones for DMs, given the state of current privacy attacks targeting the respective model types.
> > >
> > >
> > > >**However, the experimental section of the paper is robust and provides some insights for researchers. Consequently, I am inclined to give a weak accept overall.**
> > >
> > > Thank you for appreciating our empirical evaluation and maintaining the high score for our paper.

---

### Official Review · Reviewer_RYDX · 2025-03-12

**Overall Recommendation:** 2

**Summary:**

This paper presents a thorough investigation into the privacy risks of image autoregressive models (IARs), comparing them to diffusion models (DMs). The authors develop novel membership inference attacks (MIAs) and dataset inference (DI) methods tailored to IARs. Besides, they also extract hundreds of training samples from IARs. Overall, this paper is an interesting attempt of privacy attacks towards IARs.

## Update After Rebuttal

While the authors have engaged with critiques raised during the review process, the core concern regarding the paper’s central claim—that autoregressive models are inherently more vulnerable to privacy attacks than diffusion models—remains unresolved. My concerns are as follows:

​​The assertion that autoregressive architectures are "more vulnerable" to privacy attacks is misleading, as vulnerability in this context is inherently tied to the effectiveness of specific attack methodologies, not the model class itself. The rebuttal fails to provide empirical evidence isolating architectural properties as the primary factor influencing attack success rates. Here, I explain why the experiments provided in the rebuttal fails: The attack performance is associated with intrinsic model vulnerabilities. Without controlling for variables such as attack implementation, training data overlap, or model capacity, the comparison lacks rigor. A poorly tuned diffusion model could exhibit higher vulnerability under certain attacks, rendering the generalized claim untenable. This loose central claim also raises concerns about community impact. Publishing this claim without stronger empirical and theoretical grounding risks misleading the research community’s understanding of privacy risks in generative models such as DMs and IARs. The rebuttal does not address this broader implication or propose nuanced framing to mitigate potential misinterpretation.
​

**Claims And Evidence:**

The study claims that image autoregressive models (IARs) inherently exhibit heightened privacy vulnerabilities compared to diffusion models (DMs), as asserted in Lines 105–107 and empirically supported by comparative metrics in Figure 1. However, this conclusion raises questions regarding its generalizability across the broader landscape of contemporary generative architectures. Notably, the evaluation omits emerging DM variants such as flow-matching methods. Furthermore, critical factors—including model training duration and data duplication rates—significantly influence membership inference attack (MIA) efficacy. The authors should temper their generalized claim by incorporating qualifiers such as 'under the evaluated configurations' or 'potentially,' thereby aligning the conclusion more closely with the scope of empirical evidence.

**Essential References Not Discussed:**

Readers can understand the main idea of this paper given current related works.

**Experimental Designs Or Analyses:**

The experimental designs in this paper primarily follows previous arts. Thus, the designs are nothing to blame.

**Methods And Evaluation Criteria:**

The dataset in the experiments is ImageNet-1K, which may raise concerns about the scalability of the proposed privacy attacks.

**Other Comments Or Suggestions:**

See Weaknesses.

**Other Strengths And Weaknesses:**

## Strengths

1. This paper is the first to explore the privacy risks in IARs.

2. The paper explores various privacy attacks including MIA, DI and data extraction attack.

## Weaknesses

1. **The primary concern of this paper is its novelty.** Though it is the first to explore the privacy risks in IARs, it tells an old story similar in DMs and LLMs. For instance, the proposed MIA method is mainly based on CLiD conditional overfitting assumption without showing what is unique about IARs. What makes IARs privacy leakage different? Do their token generation or stacked transformers inherently make them riskier? The paper just repeats the same old "conditional overfitting" story we have heard for DMs. The authors are required to clearly explain **why** IARs are special, either in how they're build or what new risks they create. Right now, it's like saying "DMs have privacy issues... and guess what? IARs do too". That does not bring much new to the table. Highlight the difference between DMs and IARs, either from the organization of the paper or from the theoretical perspective, will help improve the contributions of this paper.

2. The target IAR models are somewhat limited. Only class-conditional models trained on ImageNet are utilized. However, most real world  concerns involve text-to-image models (e.g. copyright infringment). Evaliation on more models would actually matter for real-world harm, especailly the models trained on messy, large scale datasets like LAION.

Overall, I recognize the contribution of the paper as the first to explore the privacy attacks in IARs. However, personally, only retelling story in DMs again does not match the high standard of ICML. Therefore, I give the weak reject score.

**Questions For Authors:**

Sea Weaknesses.

**Relation To Broader Scientific Literature:**

The paper connects to prior work on MIAs for DMs and LLMs. Besides, it also transfors dataset inference in LLM and data extraction attack in DM to IARs.

**Theoretical Claims:**

N/A

---

> ### Author Rebuttal · Authors · 2025-04-01
>
> >**Emerging DMs (e.g. flow-matching)**
>
> We extend evaluation to 1) latent flow matching (LFM) (Dao et al., 2023), 2) sparse DM (DiT-MoE), 3) flow matching transformer (SiT) (Ma et al., 2024), We report:
>
> ||Model|TPR@FPR=1%|P (DI)
> -|-|-|-
> LFM|1.79|2000
> DiT-MoE|1.70|2000
> SiT|6.38|300
>
> We observe that while LFM and DiT-MoE display privacy leakage comparable to other DMs, SiT is more vulnerable. However, the leakage is still smaller than for IARs.
>
> >**Factors vs MIA efficacy**
>
> We compare training duration, model size, and a binary “Is the model an IAR?” factor against MIA and DI performance metrics, reporting Pearson’s correlation:
>
> ||Class|Duration|Size|Is IAR
> -|-|-|-|-
> P (DI)|IAR|0.24|-0.39
> P (DI)|DM|-0.58|-0.32
> P (DI)|All|-0.04|-0.28|-0.46
> |
> TPR@FPR=1%|IAR|0.17|0.93
> TPR@FPR=1%|DM|0.31|0.11
> TPR@FPR=1%|All|-0.2|0.87|0.38
>
> 1. **Duration** influences MIA/DI against DMs the most.
> 2. **Model size** influences leakage in IARs more than in DMs.
> 3. **Is IAR** factor has the strongest correlation to DI performance.
>
> We cannot isolate duplicates as a factor without re-training models from scratch. However, all evaluated models (DMs/IARs) are trained on ImageNet-1k (same duplicates).
>
> >**Temper claim**
>
> We adjust our claims to be more precise and state that the privacy risks for IARs are *empirically* more severe than DMs, given the state of current privacy attacks.
>
> >**Dataset - scalability**
>
> We acknowledge this concern, however:
>
> 1. IARs trained on >1M images (Han et al., 2024) do not specify their training data. Thus, a sound MIA/DI evaluation is impossible, as they need a) train data (members), b) IID non-members. Failure to satisfy b) leads to dataset detection (Das et al., 2024), and without a) we have no data to perform the attacks.
>
> 2. These are far from “toy models”, as ImageNet-1k allows for high quality, diverse generation.
>
> 3. The dataset is widely used as a benchmark; most cutting-edge DMs and IARs are trained on it.
>
> We believe our setting is useful for practitioners, while ensuring **full methodological correctness**.
>
> >**Novelty**
>
> We gladly clarify the novelty:
>
> 1. **First empirical privacy leakage evaluation of IARs**. We employ the strongest model-specific attacks, and perform comprehensive analysis across publicly available models.
>
> 2. **First IAR-tailored MIA**. We combine LLM-like properties of IARs with ideas from attacks against DMs to craft our MIA, improving TPR@FPR=1% by up to **69%** over the naive baseline.
>
> 3. **First IAR-specific DI**. We decrease the number of samples needed for DI for IARs by up to **90%** compared to baseline.
>
> 4. **Successful extraction attack**. We are the first to recover training data from IARs, leaking up to **698** images.
>
> 5. **Privacy-utility trade-off**. IARs are fast, but less private. Next, we explain **why**:
>
> >**What makes IARs leakage different? DMs vs. IARs**
>
> Inherent causes for higher privacy leakage in IARs:
>
> 1. **Access to p(x) boosts MIA** (Zarifzadeh et al., 2024). DMs do not expose it at inference–they learn to transform N(0,I) to data. IARs are trained to output p(x) directly. It is reflected in distinct MIA designs for DMs and IARs–the former exploit the noise, the latter–p(x), via logits. MAR does not output p(x), and is less prone to MIA (Tab. 1).
>
> 2. **AR training exposes IARs to more data per update**. RAR *outputs* 256 distinct sequences to predict a sample. DMs operate only on *a single, noised image*. At fixed training duration, leakage is stronger for IARs. VAR *outputs* 10 sequences of tokens, and is less prone than RAR to MIA (e.g., VAR-d20 vs. RAR-L of similar size).
>
> 3. **Multiple *independent* signals amplify leakage**. Each token predicted by IARs leaks a unique signal, as it is generated from a different prefix. DMs’ outputs are tightly correlated, and the aggregated signal is weaker.
>
> Architectural design choices for DMs and IARs differ for *every* model; it makes single-point conclusions unsound.
>
> >**Limited IARs. Real world: text-to-img models, on messy, large datasets**
>
> Due to the reasons highlighted in our answer on scalability, we cannot soundly evaluate larger models due to *lack of access to train data* and *lack of IID non-members*.
>
> Still, we added experiments on VAR-CLIP (Zhang et al., 2024), a text-to-img VAR trained on a captioned ImageNet-1k, reporting:
>
>
> |Model|TPR@FPR=1%|P (DI)
> -|-|-
> VAR-CLIP|6.30|60
> VAR-d16|2.18|200
> VAR-d20|5.92|40
>
> We compare VAR-CLIP to VAR-d16, as these models have the same size (300M). Notably, text-to-img model exhibits greater privacy leakage, on the level of a model twice as big, VAR-d20.
>
> >**Retelling DMs story**
>
> Our work goes beyond mirroring findings in DMs. We introduce the strongest privacy attacks available, and evaluate many public SOTA models. We empirically show that IARs are *significantly* more vulnerable than DMs; we explain **why** in the updated version of the paper. Thereby, our paper offers a valuable insight on the privacy of novel generative models.

---

> > ### Comment · Reviewer_RYDX · 2025-04-02
> >
> > Thank you for your rebuttal. I still have some questions unsolved.
> >
> > > Factors vs MIA efficacy
> >
> > Could the authors provide more detailed documentation regarding the derivation process of the Table? Specifically, I would appreciate a precise explanation of:
> >
> > 1. The detailed process to calculate the Pearson correlation coefficients, including the variables involved.
> >
> > 2. The definition of "duration size" "duration" as presented in the table. It would be beneficial to include concrete examples illustrating how these concepts are quantified.
> >
> > > Dataset - scalability
> >
> > I would like to clarify that my original comment **did not** characterize ImageNet as a "toy dataset" - in fact, I concur that models trained on this benchmark can produce visually coherent outputs. However, this observation serves to emphasize my core argument: The current privacy discourse predominantly concerns real-world deployment scenarios (e.g., user data leakage and copyright infringement) rather than theoretical vulnerabilities in research-oriented models. I comprehensively understand that it is impossible to conduct experiments on large-scale text-to-image IAR models. However, this remains a concern.
> >
> > > What makes IARs leakage different? DMs vs. IARs
> >
> > I agree with the provided causes 1 (i.e. **the access to $p(x)$ boosts MIA**). I suggest include and expand the discussion in the revised manuscript, which will provide valuable insights.
> >
> > Regarding the comparative vulnerability analysis between model architectures, I maintain the statements that the vulnerabilities requires more nuanced treatment:
> > Vulnerability should be explicitly defined through quantifiable metrics (e.g., attack success rates under standardized conditions) rather than architectural characteristics. Privacy leakage susceptibility is inherently multifactorial, depending on implementation details, training protocols rather than model architectures alone.
> >
> > > **To facilitate final evaluation of the improvements, I recommend the authors formally submit their revised manuscript incorporating the agreed-upon modifications and clarifications.**

---

> > > ### Author Response · Authors · 2025-04-04
> > >
> > > >**Factors vs MIA efficacy: derivation process of the Table.**
> > >
> > > We are happy to clarify how we obtained the results in the table. We collect five variables: TPR@FPR=1% (MIA), $P$ (DI metric), model size, duration, and *Is IAR* for every model we evaluate in the paper (11 IARs, 8 DMs). For the first two (MIA, DI) we take them directly from Tables 1, 3, 13. We obtain the model size by loading the checkpoints and summing the sizes of all the parameters in the models. (Training) duration is expressed by a number of data points passed through the model at training, e.g., for RAR-B we have 400 epochs of ImageNet-1k train set, which amounts to 400 x 1.27M≈0.5B samples seen. *Is IAR* factor is a 1 if the model is IAR, 0 otherwise. We take these variables and compute pairwise Pearson’s correlation between them, using values for all the models.
> > >
> > > >**The current privacy discourse predominantly concerns real-world deployment scenarios (e.g., user data leakage and copyright infringement) rather than theoretical vulnerabilities in research-oriented models. I comprehensively understand that it is impossible to conduct experiments on large-scale text-to-image IAR models. However, this remains a concern.**
> > >
> > >
> > > We expanded the Limitations section to accommodate these concerns.
> > >
> > >
> > > >**What makes IARs leakage different? I agree with the provided causes 1 (i.e. the access to p(x) boosts MIA). I suggest including and expanding the discussion in the revised manuscript, which will provide valuable insights.**
> > >
> > >
> > > Thank you for the acknowledgement. We included the causes and expanded the discussion in the revised manuscript.
> > >
> > >
> > > >**Regarding the comparative vulnerability analysis between model architectures, I maintain the statements that the vulnerabilities require more nuanced treatment: Vulnerability should be explicitly defined through quantifiable metrics (e.g., attack success rates under standardized conditions) rather than architectural characteristics. Privacy leakage susceptibility is inherently multifactorial, depending on implementation details, training protocols rather than model architectures alone.**
> > >
> > >
> > > We agree that having a fixed, standardized training setup for all the models would yield more reliable results. However, due to inherent discrepancies in the design and training specifics of the models such setup is infeasible. One of the reasons for that is the training objective of DMs: we train to minimize the expected error over **timesteps and data**, whereas for IARs we minimize it only **over the data**. Effectively, DMs are, *on average*, trained twice as long as IARs to match the comparative FID.
> > >
> > >
> > > We provide the fair comparison between IARs and DMs in the following way: the models we consider express the **state-of-the-art** performance given their unique architecture and training design. We compare models that are an **”upper bound”** of what is possible with inherent limitations and trade-offs each architecture has to offer. We are deeply aware that privacy vs utility is a balancing act: better models tend to be less private. Thus, our study fixes one of these parameters–utility–to be the highest possible for a given model, and under this condition we evaluate how much privacy is leaked. We believe our results provide strong empirical evidence that DMs constitute a Pareto optimum when it comes to image generation–they are comparable in FID, while being significantly more private than the novel IAR models.
> > >
> > >
> > > >**To facilitate final evaluation of the improvements, I recommend the authors formally submit their revised manuscript incorporating the agreed-upon modifications and clarifications.**
> > >
> > > We are happy to submit the updated paper (we inquired the AC regarding this option, as neither updating the submission nor providing link to extra text seems to be allowed this edition, via https://icml.cc/Conferences/2025/PeerReviewFAQ#discussions).
> > > In the current response, we have included a detailed breakdown of the changes made to the manuscript, along with corresponding line numbers:
> > >
> > > 1. We added results for emerging flow-matching DMs (Appendix).
> > >
> > > 2. We added analysis on the relation between other factors (model size, training duration) and MIA/DI performance (Appendix).
> > >
> > > 3. The claims in our paper were tempered down and we highlight the *empirical* nature of our findings (lines 39-44, 76, 78-87, 100-109, 445-458).
> > >
> > > 4. We included a section with thorough discussion about the inherent properties of IARs that increase the leakage compared to DMs (lines 327, 328, 374, 375, Appendix).
> > >
> > > 5. We incorporated the experiment on VAR-CLIP into the manuscript, with a discussion on the generalizability of our claims to broader, more messy training datasets (Appendix, Limitations).
> > >
> > > The Camera Ready version will incorporate the above changes if accepted.
> > >
> > > We thank the Reviewer for the valuable feedback and hope that our answers address all the concerns.

---

### Official Review · Reviewer_ZL49 · 2025-03-12

**Overall Recommendation:** 4

**Summary:**

The paper propose new SOTA methods for membership/dataset inference of image autoregressive models. The authors compare the privacy leakage of the different types of image generation models, and show that autoregressive models showcase important privacy leakage (up to MIA at 86.38% TPR@FPR=1%)

**Claims And Evidence:**

I did not notice problematic claims, except maybe the fact that the authors are claiming that image autoregressive models are now the gold standard for image generation, while it has not been so widely adopted.

**Essential References Not Discussed:**

Not that I am aware of .

**Experimental Designs Or Analyses:**

Yes. The experimental design is very good and well explained. However, the proposed MIA / Dataset Inference method is very succintly explained in page 5. It would have been nice to have a more detailed explanation.

**Methods And Evaluation Criteria:**

Yes.

**Other Comments Or Suggestions:**

- “IARs can achieve better performance than their DM-based
counterparts.”—> Its not that clear that IARs will take over the world
- The authors define memorization as verbatim memorization
- “We provide a potent DI method for IARs, which requires as few as 6 samples to assess dataset membership signal.”-> This depends on model size, overfitting etc. Should be clearer if its a realistic case and that comparaison is done apple to apple in terms of FID compared to diffusion models.
- The details on the MIA arrive late in the paper, in page 5
- “Interestingly, we find that t = 500 is the most discriminative, differing from the findings for fullscale DMs, for which t = 100 gives the strongest signal.” —> no figure or table to refer to for these results?

**Other Strengths And Weaknesses:**

The paper is very clear and well written, and the contribution + results are good.

As weaknesses:

1) I have found that the description of the proposed MIA/DI method is to succint and not very clear. An additional figure to explain, or equations, would have made things clearer.
2) MIA needs members and non members from the same distribution. The authors do not details how the non members are sambled which is very important in practice if one wants to do an MIA in a realistic setting.

**Questions For Authors:**

See above.

**Relation To Broader Scientific Literature:**

The paper is well positioned. To the best of my knowledge, the claim on first MIAs for image regressive models is valid. Moreover, it cites the rest of the litterature correctly.

**Theoretical Claims:**

No theoretical claim

---

> ### Author Rebuttal · Authors · 2025-03-31
>
> We thank the Reviewer for the insightful comments. We address individual points below one by one:
>
> >**[...] authors are claiming that IARs are now the gold standard for image generation, while it has not been so widely adopted.**
>
> We clarify that we position IARs as novel model family that *can* perform on par or slightly better than DMs according to the established benchmarks. Given this, we find investigating the privacy leakage of IARs at the early stage of adoption valuable for the community to support responsible adoption.
>
> >**The proposed MIA/[DI] method is very succinctly explained in page 5. It would have been nice to have a more detailed explanation. [...] I have found that the description of the proposed MIA/DI method is too succinct [...]. An additional figure to explain, or equations, would have made things clearer.**
>
> To address the Reviewer's suggestion, we further expanded and clarified our methods in the revised manuscript, including visual diagrams and procedular steps. Here, we provide a summary :
>
> 1. **MIA for VAR/RAR**: For IARs the output token probabilities are additionally conditioned, e.g., on class labels, yielding $p(x|c)$. We follow CLiD (Zhai et al., 2024) to exploit the conditional overfitting of IARs and provide $p(x|c) - p(x|c_{null})$ as input to MIAs methods (described in more detail in App. B).
>
> 2. **MIA for MAR**: we select the optimal *diffusion timestep* and *mask ratio*, perform multiple inferences (to limit the variance of the diffusion process), and obtain per-token losses per pass. We average them across inferences, and input these per-token losses to MIAs (App. B). We use losses, as logits are unavailable for MAR which outputs continuous tokens.
>
> 3. **DI improvement**: LLM DI [2] uses a scoring function $s$ to aggregate signals from the features. We note that this increases $P$ (number of samples required for DI), since a subset of samples is used to fit $s$. We replace $s$ with a summation of normalized features instead. Additionally, instead of using the original MIAs from [2], we substitute them with our improved versions (points 1. and 2 above).
>
> >**MIA needs members and non members from the same distribution. The authors do not detail how the non-members are sampled, which is very important in practice if one wants to do an MIA in a realistic setting.**
>
> We agree members and non-members *have to be* from the same distribution for the MIA/DI results to be sound. In Sec. 4, we state *“For MIA and DI, we take 10000 samples from the training set as members and also 10000 samples from the validation set as non-members.”* (lines 235-237). Because the validation set of ImageNet-1k was selected randomly from the full dataset, members and non-members are IID, which satisfies the requirement.
>
> >**“IARs can achieve better performance than their DM-based counterparts.”—> Its not that clear that IARs will take over the world**
>
> We fully understand the Reviewer’s concerns and tuned down this sentence to “IARs are an emerging competitor to DMs”.
>
> >**The authors define memorization as verbatim memorization**
>
> We explore the worst-case memorization, following the setup from [1].
>
> >**“We provide a potent DI method for IARs, which requires as few as 6 samples to assess dataset membership signal.”-> This depends on model size, overfitting etc.**
>
> Indeed, we presented the strongest result. Following the Reviewer's idea, we provide a comparison between two factors: model size and a binary “Is the model an IAR?” factor and $P$ (DI metric) as a Pearson’s correlation:
>
> ||Class|Size|Is IAR
> -|-|-|-
> P (DI)|IAR|0.24|-0.39
> P (DI)|DM|-0.58|-0.32
> P (DI)|All|-0.04|-0.28|-0.46
>
> 1. **Model size** influences leakage in DMs more than in IARs.
>
> 2. **Is IAR** is the factor with the strongest correlation to DI performance.
>
> >**Should be clearer if its a realistic case and that comparison is done apple to apple in terms of FID compared to [DMs].**
>
> Fig. 1 (left) and Fig. 2 show direct comparison between DMs and IARs in terms of FID (y-axis), where IARs exhibit greater privacy leakage than DMs, for similar values of FID. For example, in Fig. 1 (left), we observe that VAR-d24 (second blue dot from the right) has a FID of ~2.0, but the TPR@FPR=1% for this model is ~22%. In comparison, SiT achieves FID of also ~2.0, while maintaining the MIA performance of ~6% TPR@FPR=1%. We acknowledge that we do not compare privacy leakage at *a fixed FID*, but we believe these plots serve as privacy-utility trade-off curves.
>
> >**“Interestingly, we find that t = 500 is the most discriminative, differing from the findings for fullscale DMs, for which t = 100 gives the strongest signal.” —> no figure or table to refer to for these results?**
>
> We apologize for the imprecise formulation. We base the claim about fullscale DMs on [1]. We added the citation to the manuscript.
>
> **References:**
>
> [1] Carlini et al., Extracting Training Data from [DMs], USENIX 2023.
>
> [2] Maini et al.,  LLM Dataset Inference [...], NeurIPS 2024.

---

### Decision · Program_Chairs · 2025-05-01

**Decision:**

Accept (poster)

**Comment:**

This paper proposes and evaluates membership inference attacks against image autoregressive (IAR) models. Reviewers generally appreciated the novelty of studying MIA for IARs, and found the strong MIA result compared to diffusion models to be interesting and meaningful. However, reviewers also cited some weaknesses, including:
1. Missing ablation study on factors that affect memorization, e.g. number of training steps and duplicated data.
2. Findings largely agree with what is known for diffusion models and autoregressive text models without providing additional insight.
3. Only studies class-conditional models rather than text-to-image models.

The authors included additional experiments in the rebuttal aimed at addressing these weaknesses. As IARs become more prevalent and serve as efficient alternatives to diffusion models, this study is likely to have significant impact for future studies on memorization in generative models, even if the study is deemed not deep or comprehensive enough to some reviewers. For this reason, AC believes the paper's merits outweigh its weaknesses and recommends acceptance. The authors are strongly encouraged to revise the draft to include experiments and relevant discussion from the rebuttal.